# System-wide analysis of RNA and protein subcellular localization dynamics

Eneko Villanueva [1,5], Tom Smith [1,2,5], Mariavittoria Pizzinga [2,3,5], Mohamed Elzek[2], Rayner M. L. Queiroz[1], Robert F. Harvey [2], Lisa M. Breckels [1], Oliver M. Crook [4], Mie Monti[2], Veronica Dezi[2], Anne E. Willis [2] ✉ & Kathryn S. Lilley [1] ✉

Although the subcellular dynamics of RNA and proteins are key determinants of cell homeostasis, their characterization is still challenging. Here we present an integrative framework to simultaneously interrogate the dynamics of the transcriptome and proteome at subcellular resolution by combining two methods: localization of RNA (LoRNA) and a streamlined density-based localization of proteins by isotope tagging (dLOPIT) to map RNA and protein to organelles (nucleus, endoplasmic reticulum and mitochondria) and membraneless compartments (cytosol, nucleolus and cytosolic granules). Interrogating all RNA subcellular locations at once enables system-wide quantification of the proportional distribution of RNA. We obtain a cell-wide overview of localization dynamics for 31,839 transcripts and 5,314 proteins during the unfolded protein response, revealing that endoplasmic reticulum-localized transcripts are more efficiently recruited to cytosolic granules than cytosolic RNAs, and that the translation initiation factor eIF3d is key to sustaining cytoskeletal function. Overall, we provide the most comprehensive overview so far of RNA and protein subcellular localization dynamics.

Compartmentalization of eukaryotic cells and the dynamic distribution of macromolecules such as RNA and proteins across these compartments are vital for cell function. The regulation of protein production involves spatially restricted RNA–protein interactions to regulate post-transcriptional processes[1], including the creation of translation 'hotspots' so that newly synthesized proteins can act at precise locations without disturbing cellular protein homeostasis[2]. The ability to determine RNA and proteins subcellular localization, and how they relocalize upon perturbation, is thus key to understanding cellular homeostasis[3,4]. While cell-wide methods to study protein localization are well established[5–7], equivalent methods to study RNA subcellular localization are limited.

The localization of a single RNA transcript can be determined using single molecule fluorescence in situ hybridization (smFISH)[8].

Alternatively, the RNA content of specific niches can be explored using proximity-dependent biotinylation techniques, such as APEX-RIP[9] and APEX-seq[10,11], or by multiplexing smFISH in combination with immunofluorescence (IF) for a localization marker[12]. However, all these approaches are restricted to the interrogation of a single subcellular compartment per experiment and thus cannot provide a cell-wide view of RNA localization. Combining multiple APEX experiments to generate a more complete understanding of RNA localization is time-consuming and does not quantify the proportion of RNAs at each localization. Previous attempts to determine the complete localization of RNA in a cell-wide manner have been restricted by technical limitations, including localization-independent RNA clustering and length-dependent RNA localization biases[13–15]. Therefore, there is a need to develop robust methods to obtain comprehensive RNA subcellular localization

[1]Cambridge Centre for Proteomics, Department of Biochemistry, University of Cambridge, Cambridge, UK. [2]MRC Toxicology Unit, University of Cambridge, Cambridge, UK. [3]Structural Biology Research Centre, Human Technopole, Milan, Italy. [4]Department of Statistics, University of Oxford, Oxford, UK. [5]These authors contributed equally: Eneko Villanueva, Tom Smith, Mariavittoria Pizzinga. ✉e-mail: aew80@cam.ac.uk; k.s.lilley@bioc.cam.ac.uk

information, ideally in a framework that also enables determination of protein subcellular localization, so that the two biomolecules can be studied simultaneously.

In this Article, we develop such a framework that returns simultaneous cell wide spatial overviews of the transcriptome and proteome. To enable the creation of this framework, we have firstly developed a transformative method to study cell-wide subcellular localization of RNA (termed LoRNA). LoRNA builds on the principle of reduction and reconstruction of the cellular content[16] to study RNA localization. It involves reordering the subcellular constituents by their density, quantification of RNA abundance profiles across the density fractions, and using localization-specific RNA correlation profiles to determine the subcellular localization of the complete transcriptome. Importantly, unlike previous methods that estimate localization-specific enrichments, LoRNA allows the estimation of RNA proportions in each localization, providing more biologically meaningful information. Secondly, we have engineered LoRNA to allow the simultaneous interrogation of the subcellular proteome, using a novel streamlined localization of proteins by isotope tagging (LOPIT[5,6]) approach, which we refer to as density-based LOPIT (dLOPIT).

This integrative approach for the cell-wide quantification of RNA and protein distribution is especially valuable to study molecular relocalization processes. Therefore, we chose to examine the redistribution of RNA and protein during the activation of the unfolded protein response (UPR). The UPR is triggered by the accumulation of unfolded proteins in the endoplasmic reticulum (ER) lumen[17]. Its activation reduces global protein synthesis rates, resulting in the loss of RNA targeting to the ER, the formation of stress granules (SGs), and the upregulation of stress-response genes[18]. Importantly, UPR deregulation is associated with disease states including neurodegeneration[19], cancer progression[20] and diabetes[21]. Combining LoRNA and dLOPIT, we simultaneously quantified the extensive reorganization of the transcriptome and proteome upon the activation of the UPR and resultant inhibition of translation, including the loss of RNAs from the ER and the relocalizations of proteins from the secretory pathway. Our integrative approach enabled the discovery of cytosolic granules in our multiomic framework and reveals that ER-localized RNAs are recruited to granules to a greater extent than previously thought[22–24]. Our data also show that RNAs encoding cytoskeletal proteins are retained and targeted to the periphery of organelles during UPR and that this process involves eIF3d. Importantly, while messenger RNAs undergo a profound subcellular reorganization upon UPR activation, noncoding RNAs sharing the same properties (for example, size and nucleotide composition) do not, indicating the importance of *trans* factors in determining RNA localization. Altogether, this uniquely integrative approach has allowed us to generate the most comprehensive analysis of RNA and protein cell-wide localization dynamics so far, which can be readily explored using dedicated open-source resources[25,26].

## Results

### Cell-wide RNA and protein subcellular localization

Organelles can be separated on the basis of their physicochemical properties to determine the localization of macromolecules[5,27,28]. Unfortunately, the application of this principle to interrogate RNA has so far failed to capture the complete subcellular distribution of RNA[13,14]. To resolve this issue, we developed a new approach to sort the entire cellular content on the basis of the density of its constituents (Fig. 1a). We applied this framework to U-2 OS cells. Cells were first mechanically lysed with a ball-bearing homogenizer, using a 12-μm pore to lyse the cells while preserving the integrity of the intracellular organelles. The complete cell lysate was then loaded into a precast equilibrium gradient (1 to 1.6 g ml⁻¹, see experimental details). Crucially, unlike previous methods that use density gradients to resolve subcellular compartments[13], the cell lysate was loaded at the density of free RNA (1.17 g ml⁻¹). This ensured that organelle-associated RNAs migrated

upwards in the density gradient, while cytosolic RNA–protein complexes migrated downwards, allowing free RNA and organelles to reach equilibrium faster and avoiding cross-contamination between cellular compartments (Supplementary Fig. 1a). After 16 h of centrifugation at 100,000g, three distinct bands were observed in the gradient, which were enriched in (1) ER and mitochondria, (2) nucleus and (3) cytosolic proteins, respectively (Supplementary Fig. 1b,c).

To determine whether this subcellular fractionation approach allowed analysis of the subcellular localization of RNA, we performed RNA sequencing (RNA-seq) along the gradient and evaluated the sedimentation profile of gene products known to localize to specific subcellular compartments based on previous targeted RNA localization experiments[11,29,30] and known nuclear and cytoplasmic long noncoding RNAs (lncRNAs) (see experimental details). Importantly, when analyzing correlation profiles, the fractions obtained are not purified organelle/compartments; distinct sedimentation profiles for each subcellular niche are required to determine localization. We observed distinct profiles resulting in separate clustering within the data for RNAs known to localize to the mitochondria, ER, nucleolus, nucleus and cytosol (Fig. 1b–d Supplementary Fig. 1d and Supplementary Tables 1 and 2). Further to these expected profiles, a novel subcytosolic RNA sedimentation profile was discovered by semi-supervised clustering. The RNAs of this profile are enriched at a density higher than the organelles, but lower than the cytosolic ribosomes (Fig. 1b and Supplementary Fig. 1d–f). Thus, we refer to this profile as 'cytosol light'. To our knowledge, this is the first time the complete subcellular localization of RNA has been successfully resolved at a cell-wide level. We refer to this method to interrogate the localization of RNA as LoRNA.

Next, we evaluated if this fractionation approach was compatible with the simultaneous analysis of the proteome localization, using LOPIT analysis methods[31]. We extracted protein from the same fractions used for LoRNA, and quantified the protein abundance by mass spectrometry (MS) (Fig. 1e). To assess the accuracy of our protein localization, we used a support vector machine to classify marker proteins for all major compartments on the basis of their abundance profiles. F1 scores (the harmonic mean of precision and recall) of 0.71–1 were observed, demonstrating high resolution of protein subcellular localizations, based purely on the density of the different compartments (dLOPIT; Fig. 1f,g and Supplementary Fig. 1g). Remarkably, dLOPIT performs comparably to our previous high-resolution LOPITs[5,6] that were engineered to exclusively interrogate the protein subcellular localization (Supplementary Fig. 1h). These results demonstrate that this integrative framework allows, for the first time, the simultaneous analysis of the subcellular localization of the transcriptome and proteome.

**Proportional quantification of RNA localization.** Relative RNA abundances were adjusted with respect to spike-in RNAs to account for the total RNA content per fraction and obtain profiles of absolute RNA abundance. This allowed the deconvolution of the RNA profiles into the constituent contributions from each localization, as determined from the profiles of RNAs with known localizations (Fig. 2a). In this way, we estimated the proportional localization of 31,839 transcript isoforms (13,142 genes) in every localization simultaneously. Additionally, transcript-level RNA localization analysis retains specific information on the differential localization of splicing variants (Supplementary Fig. 2a). To account for splice isoform localization variability, we use transcript-level proportions, unless specified otherwise. Importantly, we took advantage of the extreme localization of the 13 mitochondrially encoded mRNAs to accurately quantify transcript membrane association, using these RNAs as a reference for 100% membrane localization. As anticipated, RNA is distributed along two major axes: nucleus:cytosol and cytosol:membrane, with expected proportions obtained for RNAs known to be extremely enriched in specific locations (Fig. 2b).

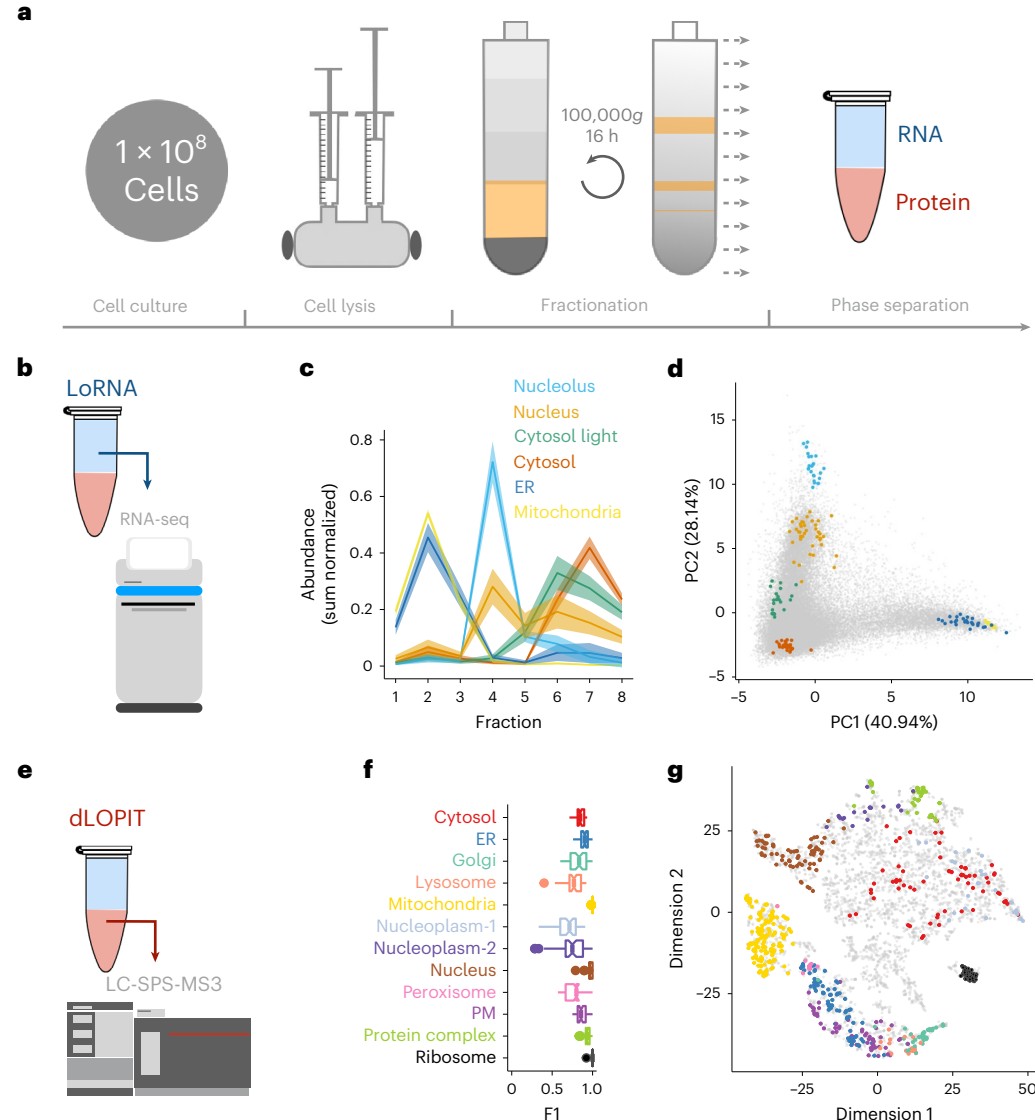

**Fig. 1 | Simultaneous analysis of RNA and protein subcellular localization.**
**a**, Schematic representation of the subcellular fractionation framework.
Cells are lysed and fractionated by density equilibrium centrifugation.
RNA and protein are extracted from each fraction to perform LoRNA and
dLOPIT. Cell lysate and the gradient banding pattern are represented in yellow.
Aqueous and organic phases are blue and red, respectively. **b**, Application of
LoRNA to U-2 OS cells. **c**, Mean profiles for RNA markers along pooled gradient
fractions in a single experiment. Shaded regions denote ± one standard error.

**d**, Principal component analysis projection of RNA profiles across three
replicate experiments, with marker RNAs highlighted. **e**, Application of
dLOPIT to U-2 OS cells. **f**, Distributions of F1 scores for protein markers for each
localization using support vector machine classification. $n = 50$ iterations. PM,
plasma membrane. **g**, t-Distributed stochastic neighbor embedding projections
for protein localization profiles across three replicate experiments, with marker
proteins highlighted.

To comprehensively validate our system-wide RNA localiza-
tion results, we developed an orthogonal method to sort the cel-
lular content by a different physicochemical property, namely the
sedimentation coefficient of the different organelles instead of
density. Previous attempts to study RNA localization using this
concept applied high $g$ forces[14,32], and analysis of the published
data highlights that this creates an RNA length-dependent localiza-
tion bias; longer RNAs precipitate with organelles and are depleted
from the cytoplasm (Supplementary Fig. 2b,c). We developed a
fractionation approach that separates the major subcellular locali-
zations while avoiding the RNA length-dependent sedimentation
bias. (Fig. 2c, Supplementary Fig. 2b–d and Supplementary Tables 1
and 2). While the low $g$ force required to avoid this bias precludes
subcytosolic resolution, differential centrifugation provides better
separation between mitochondria and ER than density centrifugation

(Fig. 2d). To assess the agreement between our two orthogonal
approaches, we projected the density-based RNA proportions
onto the sedimentation-based data. Notably, the two fractionation
approaches showed a remarkable agreement for membranes, nucleus
and cytosol RNAs (Fig. 2e). In addition to the excellent separation of
the main RNA localization niches, reliable estimates of cytosol and
membrane proportions were also achieved (Fig. 2f and Supplemen-
tary Fig. 2e). These proportions show Pearson's correlations of 0.77
and 0.83, respectively, confirming the reliability of our estimates.
Importantly, LoRNA provides accurate estimates, even when com-
pared with targeted single-localization enrichment techniques,
including APEX-seq and IF-aided multiplexed error-robust fluores-
cence in situ hybridization (MERFISH; Supplementary Fig. 2f–i).
Thus, LoRNA provides an accurate quantitative system-wide over-
view of RNA localization.

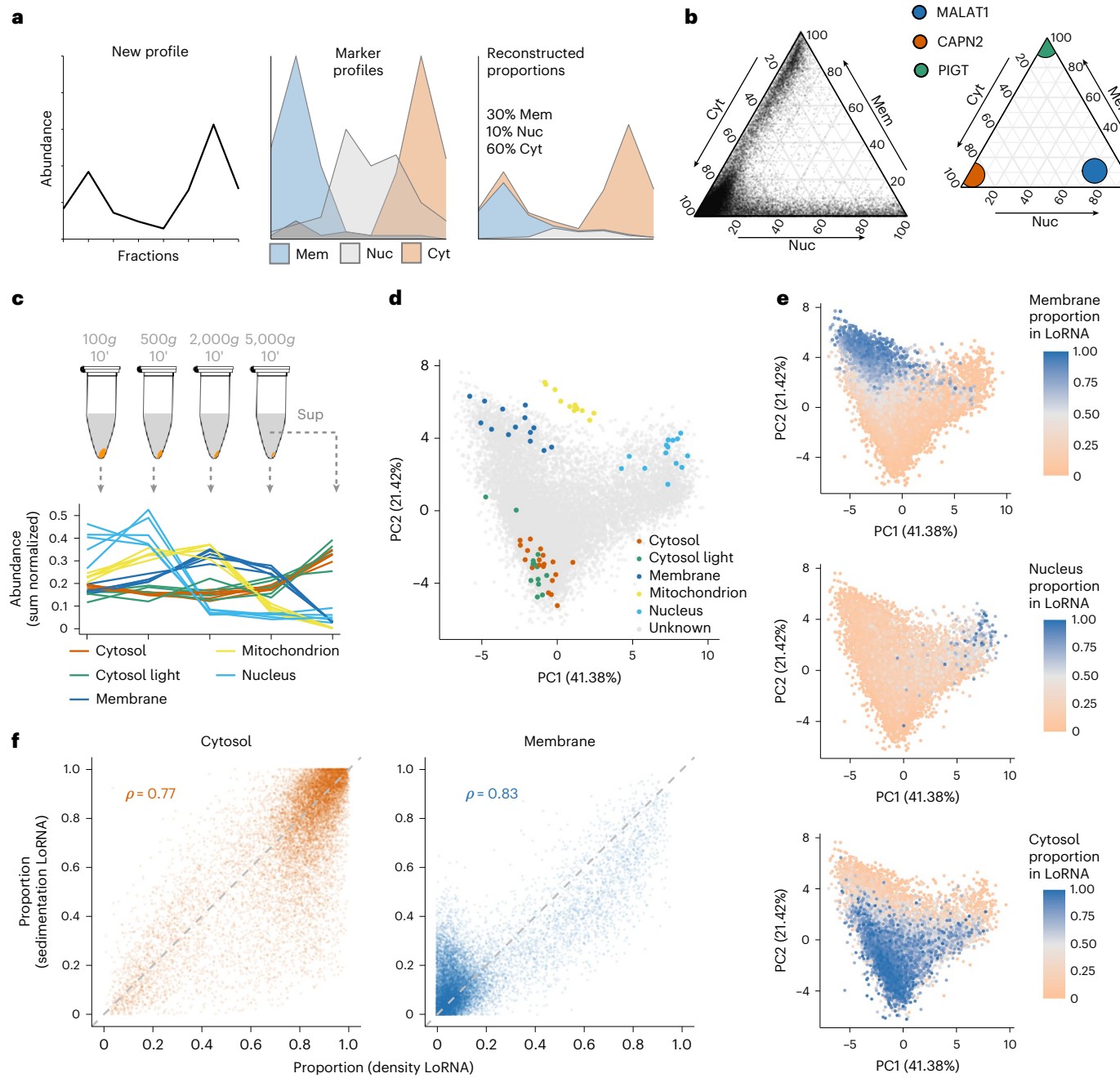

**Fig. 2 | System-wide quantification of RNA localization. a**, Schematic representation of how localization proportions are estimated. New profile (left) is decomposed into different proportions of each marker profile (middle) to approximate the original profile (right). **b**, Cytosol (Cyt), membrane (Mem) and nuclear (including nucleolus) (Nuc) proportions for all transcripts (left) and well-documented localization markers (right). **c**, Schematic representation of cell fractionation by sedimentation coefficient and linear profiles of the localization markers. **d**, Principal component analysis projections of RNA profiles, with markers highlighted. **e**, Membrane, nucleus and cytosol proportions obtained by equilibrium density centrifugation-based LoRNA projected on the sedimentation-based RNA localization results. **f**, Correlations between proportions obtained using different fractionation approaches. Cytosol proportion by density is the sum of cytosol and cytosol-light proportions. Membrane proportion by sedimentation is the sum of mitochondrial and ER proportions.

**Key RNA features drive subcellular localization.** We next explored the RNA features associated with proportional localization. While mRNAs are observed throughout the cell, and relatively depleted from the nucleus, lncRNAs were observed to be strictly within the nucleus:cytosol axis (Fig. 3a). Many lncRNAs have well-described nuclear functions for example as scaffolds for histone modification complexes[29], and nuclear/cytoplasm fractionation RNA-seq experiments indicate lncRNAs are relatively enriched in the nucleus[29]. As expected, in our dataset lncRNAs were more nuclear localized than mRNAs[33]. However, in line with a recent re-analysis of fractionation RNA-seq that accounted for the total RNA content per fraction[34], our quantification of localization proportions indicates that lncRNAs are still predominantly cytosolic (Fig. 3b). A comparison of the proportions between our two cell fractionation approaches demonstrated that 59.7% of lncRNAs were consistently cytosolic. It has been shown previously that some lncRNAs have coding potential[35]. Consistently, we identified that ribosome association correlates with greater lncRNA cytosol localization[36] (Supplementary Fig. 3a). Importantly, we found

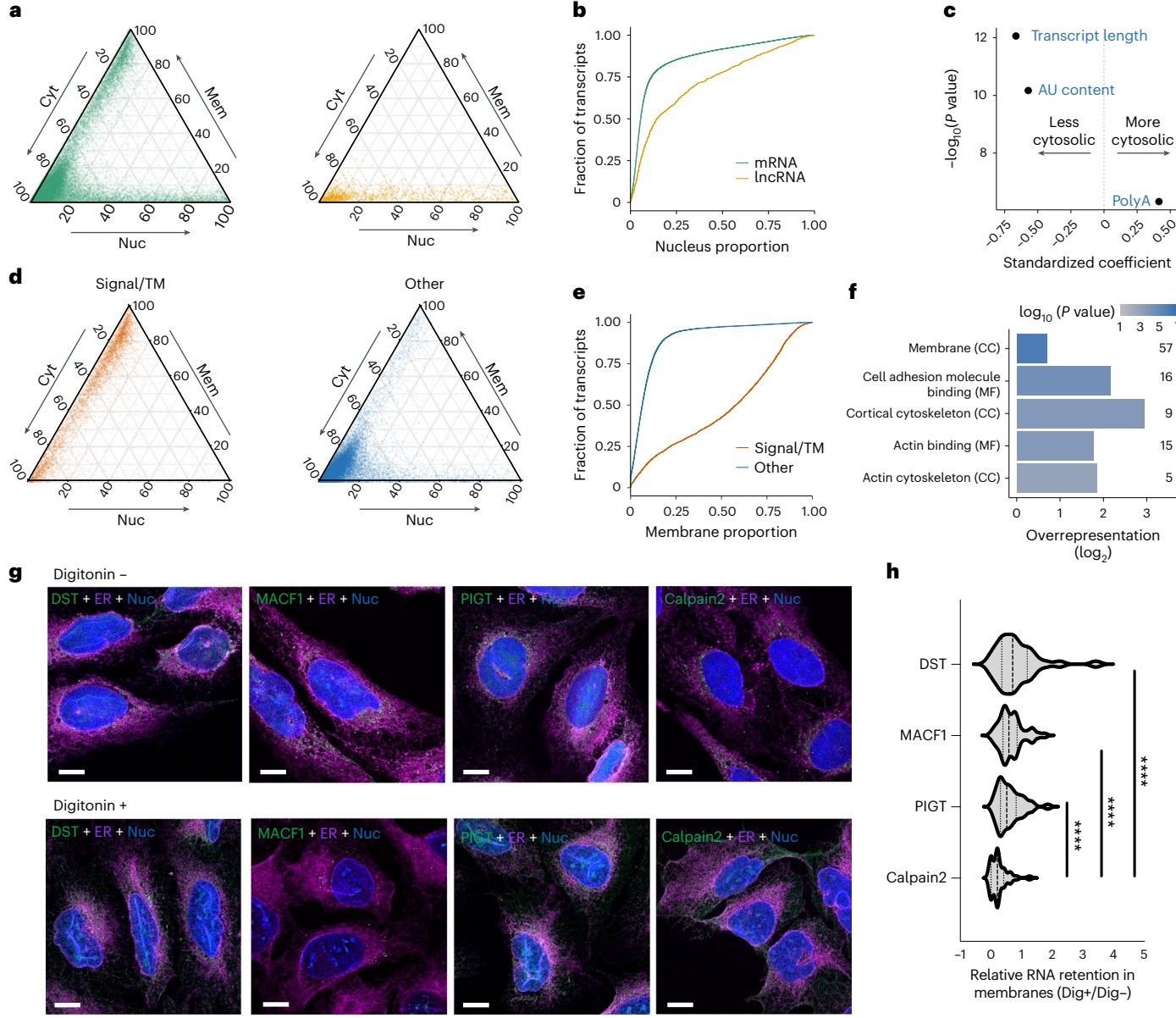

**Fig. 3 | Features driving RNA localization. a**, Cytosol (Cyt), nucleus (Nuc) and membrane (Mem) proportions for mRNAs and lncRNAs. **b**, Empirical cumulative frequency distributions for nucleus proportions for mRNAs and lncRNAs. **c**, Coefficient estimates for logistic regression model of lncRNA cytosol proportions. $P$ values derived from two-tailed $z$-test. **d**, Proportions for mRNAs encoding signal peptides and/or TM domains. **e**, Membrane proportions for mRNAs shown in **d**. **f**, GO terms significantly enriched in mRNAs not encoding signal peptides and/or TM domains but over 35% membrane localized. $P$ values derived from a two-tailed hypergeometric test and adjusted for multiple testing using the Benjamini–Hochberg FDR correction. CC, cell compartment. MF, molecular function. **g**, smFISH/IF images of digitonin-extracted (Dig+) and control (Dig−) U-2 OS cells co-stained for the ER (calnexin, magenta), and MACF1 and DST RNAs (green). Calpain2 and PIGT RNAs are shown as examples of cytosolic transcripts and known ER transcripts, respectively. Scale bar, 10 μm. **h**, Quantification of RNA transcripts in digitonin-treated cells, using control (untreated) cells for normalization. At least 30 cells were quantified per condition. ****$P < 0.0001$ (two-tailed Mann–Whitney $U$ test).

that cytosolic lncRNAs are generally shorter, have a lower AU content, and are more likely to be polyadenylated, compared with nuclear lncRNAs (Fig. 3c). A simple model using these features is as accurate as a more complex model built from a penalized regression over a wide range of potential features (receiver operating characteristic area under curve 0.86 for both models; Supplementary Fig. 3b). We therefore suggest that these are the key features driving cytosolic localization for lncRNAs (Fig. 3c).

While generally we found mRNAs predominantly in the cytosol, RNAs encoding proteins with signal peptides and/or transmembrane (TM) domains were much more prominently membrane localized,

consistently with their co-translational targeting to the ER (Fig. 3d,e). Furthermore, our data confirmed the relationship between the distance from the first signal peptide/TM domain to the stop codon and the membrane proportion (Supplementary Fig. 3c)[30,37]. Surprisingly, we also identified 87 membrane-localized mRNAs that did not encode a signal peptide or TM (Supplementary Table 2). These included the mRNA encoding the ER-localized signal recognition receptor subunit β, which co-translationally binds to the α subunit[38], and mRNAs encoding for mitochondrial proteins (Supplementary Fig. 3d), but most of the RNAs were not previously described as membrane localized. Gene Ontology (GO) analysis revealed a significant enrichment for terms

associated with membrane or cytoskeleton localization (Fig. 3f), suggesting potential localized translation at the surface of the membranes. To confirm ER association of these RNAs, we combined smFISH/IF with a digitonin-based cytosol extraction approach, focusing on two mRNAs that do not encode signal peptides or TM domains, Microtubule-actin cross-linking factor 1 (MACF1) and Dystonin (DST) RNAs. Both mRNAs are only marginally affected by digitonin treatment, similarly to known ER-associated transcript PIGT (Fig. 3g,h). By contrast, cytosolic transcript Calpain2 is strongly affected by the treatment. These results are in agreement with LoRNA data and indicate that LoRNA can accurately uncover novel, noncanonical ER-associated transcripts.

**Assessing global transcriptome and proteome relocalization.** System-wide RNA localization is especially suited for studying RNA redistribution upon stimulation, where transcripts can migrate to unexpected localizations. As a showcase, we used LoRNA to interrogate RNA relocalization upon activation of the UPR, which was induced via inhibition of the SERCA $Ca^{2+}$ pump by treating U-2 OS cells with 250 nM of thapsigargin (TG) for 1 h. This reduces calcium in the ER lumen, impairing protein folding and activating the UPR, which, in turn, rapidly induces a global translational shutdown through phosphorylation of eIF2α (Supplementary Fig. 4a,b). This results in the formation of SGs (Supplementary Fig. 4c) and the increased expression of UPR genes including XBP1 and CHOP (Supplementary Fig. 4d). LoRNA allows quantification of the cell-wide redistribution of RNA upon inhibition of translation, clearly showing a profound RNA migration from the membranes towards the cytosol (Fig. 4a and Supplementary Fig. 4e). Surprisingly, we found a pronounced relocalization of RNAs to the 'cytosol light', both from the cytosol and from membranes (Fig. 4b). We therefore hypothesized that the cytosol light profile may represent RNP granules.

To test this hypothesis and gain a deeper understanding of the cellular reorganization upon UPR activation, we exploited our fractionation approach to yield simultaneous data on proteome relocalization by applying dLOPIT to the same gradient fractions that had been used for LoRNA. A Bayesian analysis framework[39] was applied to identify protein relocalization 1 h post UPR activation, at three levels of confidence depending on the differential localization probabilities, 'highly confident' (>99%), 'confident' (>95%) and 'candidate' (>85%). This identified 73 proteins that were differentially localized with confidence (Fig. 4c,d and Supplementary Table 3). The Golgi apparatus was the most affected organelle, as expected given its role in the trafficking of newly synthesized proteins. Interestingly, four SG proteins (STAU2, UPF1, PABPC1 and PABPC) relocalized away from the ribosomes towards the fraction that differentiates the cytosol light (Fig. 4e and Supplementary Fig. 4f). We used these 4 proteins to identify a further 22 proteins with correlated UPR profiles, including 5 members of the P-body associated CCR4–NOT complex[40], P-body proteins DCP1A/B[41] and SG components YBX3 (ref. 42), SECISBP2 (ref. 42) and CASC3 (ref. [43]), further suggesting that the cytosol-light profile represents cytosolic granules. To confirm that cytosolic granules are enriched in the fractions discriminating the cytosol-light profile, we engineered U-2 OS cells to express green fluorescent protein (GFP)-tagged DCP2 and G3BP1 proteins. DCP2 is a canonical P-body protein[44], while G3BP1 is a well-known constituent of SGs[45]. By performing density-based cell fractionation experiments on cells expressing these two engineered proteins and imaging them along the gradient we confirmed that cytosolic granules sediment at the density ranges of 'cytosol light' (Supplementary Fig. 4g,h).

**RNA features driving granule localization.** The proteins relocalizing to the cytosol light upon UPR are associated with RNA condensation into granules; therefore, we further characterized the RNA composition of this fraction and interrogated the features of the enriched RNAs. RNAs with higher cytosol light abundance in unstressed cells

were correlated with lower ribosome association and longer transcript length (Supplementary Fig. 5a,b), both features associated with RNAs partitioning to granules[46,47]. Furthermore, RNAs enriched in P-bodies[48] and TIS granules[49] using fluorescence-activated particle sorting have a similar distribution to the cytosol light profile (Supplementary Fig. 5c,d). Finally, we observed a positive correlation between the cytosol light proportions upon UPR activation and SG enrichment upon arsenite treatment using targeted SG purification[23] (Supplementary Fig. 5e). Altogether, these data provide strong evidence that our cytosol light profiles represent cytosolic ribonucleoprotein-containing granules, and we henceforth refer to this localization as 'granule'.

As expected, mRNAs are recruited to granules upon UPR activation in a length and AU content-dependent manner. Surprisingly, lncRNAs do not relocalize to granules, irrespective of these features (Fig. 5a–c). Importantly, LoRNA provides cell-wide quantification of RNA localization before and after stimulation. This allowed us to uncover that, upon activation of the UPR, although many cytosolic mRNAs migrate to granules, membrane mRNAs do so at a higher proportion (Fig. 5d). However, not all membrane mRNAs relocalize to granules (Fig. 4b and Supplementary Fig. 5f). The recruitment of specific mRNAs to granules was confirmed with smFISH (Fig. 5e) and agrees with a recent targeted study of the relocalization of six RNAs upon UPR induction (Supplementary Fig. 5g)[50]. To fully characterize the features driving mRNAs towards granules upon UPR activation, we modeled the contribution of a broad range of RNA features, including transcript length, localization before stress, RNA-binding protein (RBP) binding (from enhanced cross-linking and immunoprecipitation (eCLIP) data[51]), k-mer content, codon usage and presence of internal ribosome entry site (IRES) or upstream open reading frames (uORFs). Overall transcript length, especially longer 3' untranslated regions (UTRs) in membrane mRNAs, was observed to be the greatest positive predictor of relocalization to granules (Fig. 5f). RBPs whose binding according to eCLIP was positively predictive of RNA relocalization to granules included the granule proteins FAM120A, IGF2BP1 and G3BP1 (Fig. 5g). Moreover, the binding of canonical SG proteins TAIL1, TIA1 and IGFBP3 to membrane-localized mRNAs also showed a significant predictive value. Intriguingly, unexpected associations between RBP binding and relocalization to granules were observed, including ZNF622, which is involved in ribosome subunit joining[52] and ribosome stalling[53], and upregulated upon viral infection and UPR stress[54]. Importantly, ZNF622 was a 'candidate' relocalizing protein, with movement from the cytosol to the ER upon UPR (Supplementary Fig. 5h,i), a step that may be required for the membrane-RNA recruitment to granules.

A higher AU content in the coding and 3' UTR regions was predictive of greater relocalization of mRNAs to granules (Fig. 5h). The latter is consistent with the known role of 3' UTR AU-rich elements in regulating RNA stability. Interestingly, higher G content, but not C content, in the coding region was a predictor of lower granule relocalization (Fig. 5h). In addition, a higher frequency of AUG k-mers in the 5' UTR was predictive of lower relocalization, although presence of annotated uORFs was not. This may indicate that the presence of an AUG at the 5' UTR inhibits granule association, regardless of the ORF capacity of the sequence. Finally, while overall codon optimality was not a selected predictive feature, the frequency of UUA, GAU, AAU and CAC codons were predictive of granule relocalization (Fig. 5i). Three of these are NAC/U codons, decoded by queuosine-modified transfer RNAs (tRNAs). Notably, the two NAU codons are predictive of greater relocalization to granules and the NAC codon predictive of lower relocalization, pointing to a potential role of this modification in the stress response and RNA recruitment to cytosolic granules.

**RNAs retaining organelle association upon UPR.** Despite the global loss of RNA from the membranes, some RNAs remain membrane-associated upon UPR activation (Fig. 6a). We modeled this nonlinear relationship using a generalized additive model (GAM) with

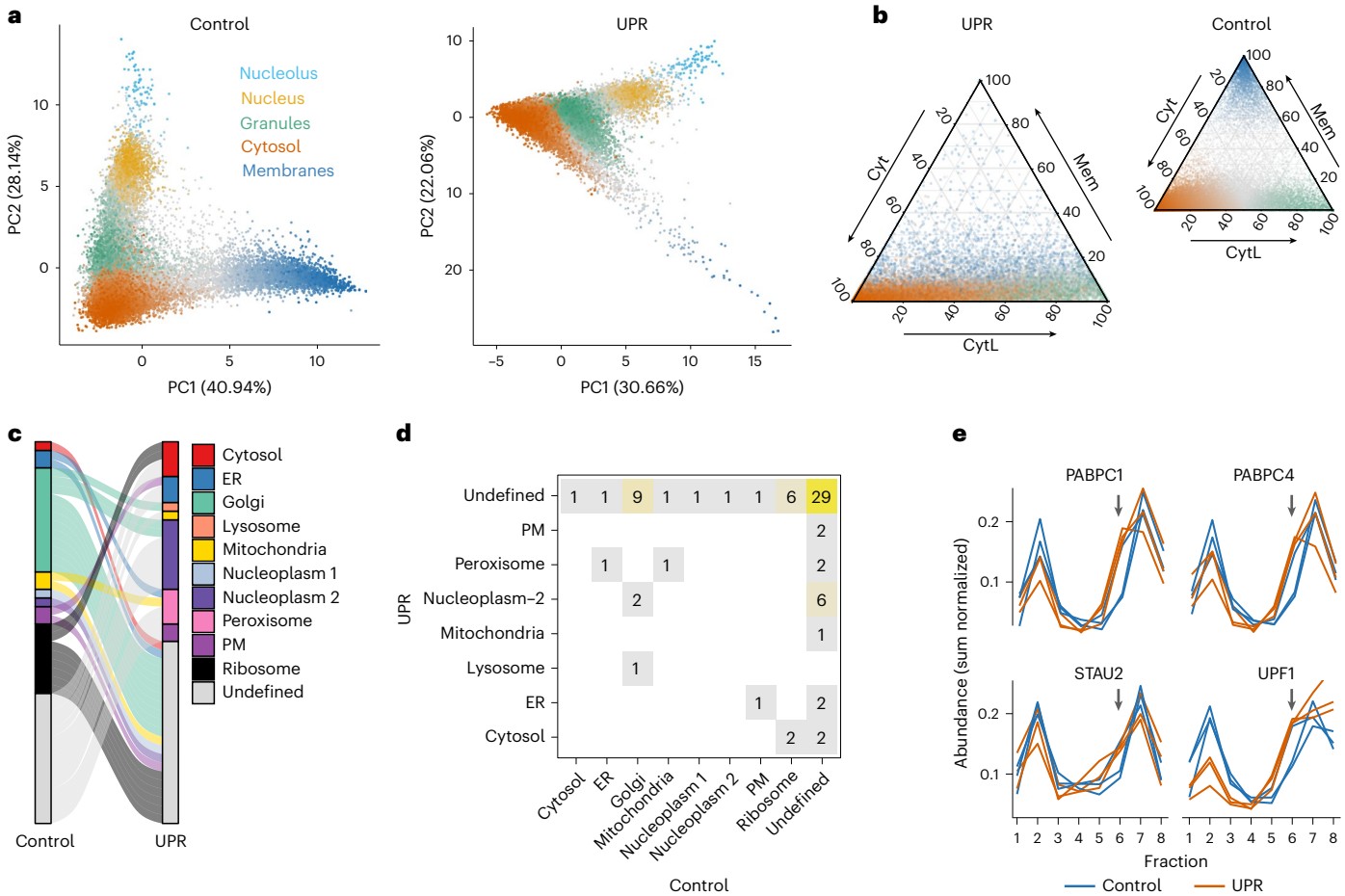

**Fig. 4 | Transcriptome and proteome subcellular redistribution upon UPR.**
**a**, Principal component analysis projection of RNA profiles in control and UPR. Color and intensity indicate RNA proportions for the primary localization. PC2 in UPR is inverted. **b**, Cytosol (Cyt), membrane (Mem) and cytosol light (CytL) proportions in control and UPR. Point color indicates RNA major localization (excluding nucleus and nucleolus) in control, with color intensity denoting proportion. **c**, Localizations for proteins with confident differential localization between control and UPR, excluding Undefined to Undefined. PM, plasma membrane. **d**, Differential localizations frequencies between control and UPR. **e**, Abundance profiles for SG proteins relocalizing from ribosomes to the fraction discriminating the cytosol-light-RNA profile under UPR; gray arrows mark cytosol light discriminating fraction.

a cubic regression spline and used the residuals from the GAM to identify RNAs with unexpectedly high membrane localization after the induction of the UPR. Notably, RNAs with longer sequences between the signal sequence and stop codon are retained more efficiently in membranes, suggesting the involvement of the signal recognition particle for these RNAs[37] and therefore active reassociation with the membranes (Fig. 6b). However, many of the membrane-localized RNAs that do not encode a TM or signal peptide (Fig. 3) were retained at the membrane upon UPR activation (Supplementary Table 2), indicating a signal recognition particle-independent mechanism for their retention. To gain an understanding of the mechanisms of RNA retention in the periphery of organelles upon UPR activation, we investigated protein–RNA interactions using publicly available eCLIP data[51]. Applying a lasso penalization model to select the proteins whose binding predicts RNA localization, we observed that the binding of 11 out of 177 RBPs was predictive of greater membrane localization, including eIF3h and eIF3d (Fig. 6c and Supplementary Fig. 6a). Surprisingly, genes whose translation is affected by the knockdown of the core eIF3 component eIF3e[55], do not show a differential membrane retention upon UPR activation (Supplementary Fig. 6b,c), suggesting that membrane retention may be specific to specialized eIF3 components. Interestingly, eIF3d has been found to maintain translation upon ER stress[56] and can directly bind at the 5′ cap[57]. We confirmed that RNAs known to be bound by eIF3d at

their 5′ cap[58] (Fig. 6d) are more retained in the membranes upon UPR activation, indicating that 5′ cap binding of eIF3d may contribute to the maintenance of this localization. Membrane-localized RNAs bound by eIF3d show an overrepresentation for GO terms relating to the actin cytoskeleton, cell morphogenesis and focal adhesions (Fig. 6e). These membrane-localized eIF3d-bound RNAs include two cytoskeletal protein coding transcripts, MACF1 and DST transcripts, which we previously showed are among the cytoskeletal-protein encoding transcripts unexpectedly associated with the ER (Fig. 3g). Using detergent-based subcellular fractionation, we further validated that the membrane localization of MACF1 and DST is eIF3d dependent (Fig. 6f and Supplementary Fig. 6e,f).

Finally, we hypothesized that the retention of eIF3d-bound transcripts in the ER upon UPR induction may be required to remodel the cytoskeleton in processes that depend on localized translation, such as cell migration, particularly during activation of the UPR. In agreement with this, specifically decreasing eIF3d expression reduces cell migration overall, with a significantly stronger effect in cells undergoing UPR (Fig. 6g,h and Supplementary Fig. 6g–j). While eIF3d has been shown to bind cytoskeletal RNAs[58] and cytoskeleton remodeling is required for calcium homeostasis upon UPR induction[59], to our knowledge, this is the first time these two processes have been connected as potentially being part of the same integrated response. Altogether, our results suggest that eIF3d is required for continued translation of actin

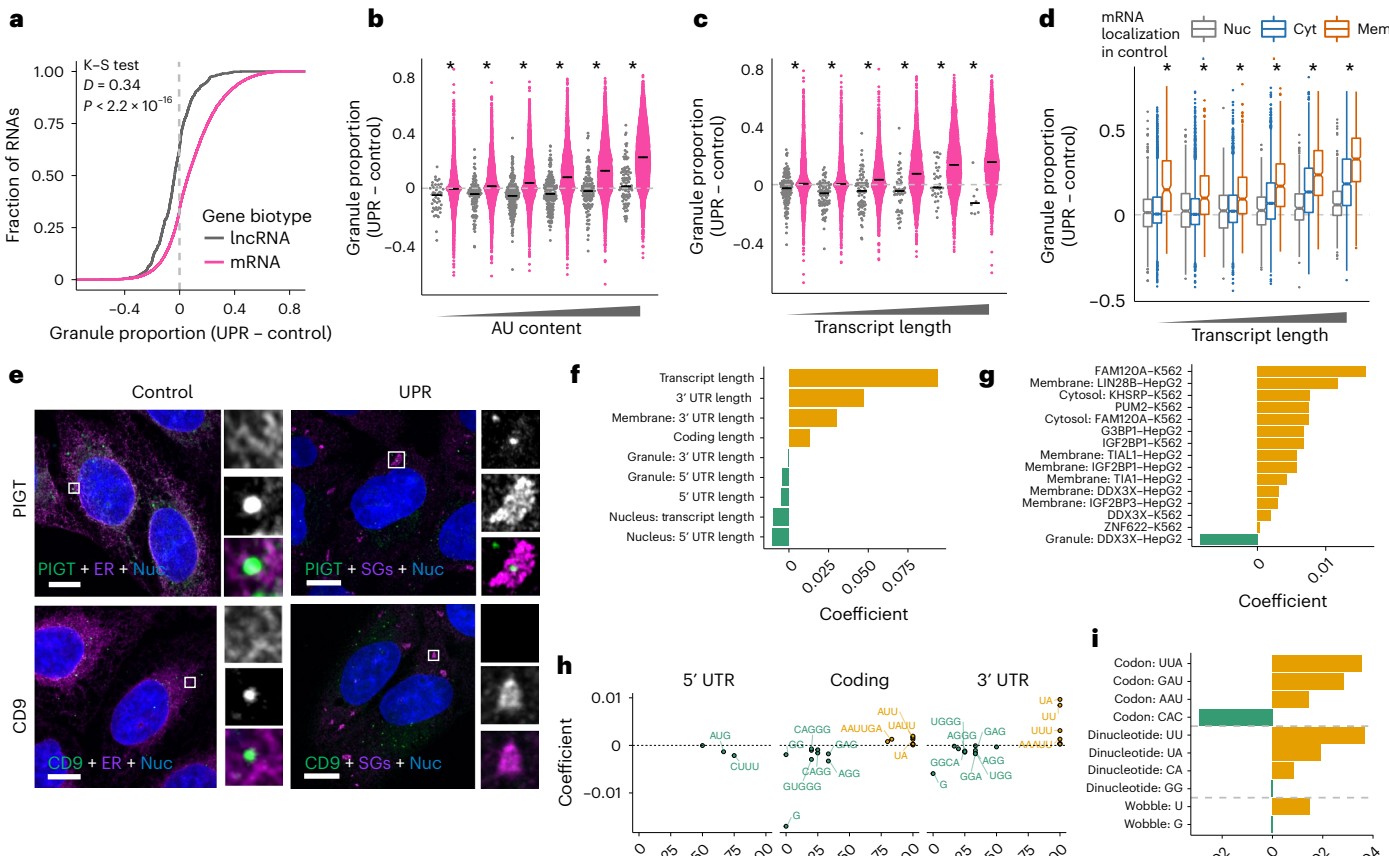

**Fig. 5 | Analysis of the characteristics driving RNAs to granules.**
**a**, Distributions for changes in granule proportions upon UPR activation for lncRNA and mRNAs. Results from a two-sample Kolmogorov–Smirnov (K–S) test that the distributions are different are shown. D, distance statistic. **b**, Relocalization to granules upon UPR activation. RNAs are binned by transcript length and split by gene biotype. lncRNAs and mRNAs are represented as gray and pink dots respectively. Black bar, median. *$P < 0.001$ two-tailed Wilcoxon rank-sum test for lncRNAs versus mRNAs. For exact $P$ values and $n$ numbers, see Supplementary Table 4. **c**, As per **b**, with bins by AU content. **d**, Relocalization to granules upon UPR activation. RNAs are binned by transcript length and split by localization in control condition. Box extends to the 25th and 75th percentiles. Whiskers extend to range, excluding outliers. Outliers are defined as greater

than 1.5× the interquartile range from the box. *$P < 0.001$ two-tailed Wilcoxon rank-sum test for membrane versus cytosol. For exact $P$ values and $n$ numbers, see Supplementary Table 4. **e**, Representative smFISH/IF images showing PIGT and CD9 RNA localization relative to the ER (calnexin) in control conditions and relative to SGs (G3BP1) upon UPR activation, $n = 3$ independent experiments. Scale bar, 10 μm. **f**, Coefficients for features selected by lasso regression model of mRNA relocalization to granules. Coefficients for RNA length features. **g**, Coefficients for features describing RBP binding or the interaction between RBP binding and localization in control. RBP eCLIP cell line indicated in the feature name. Positive coefficients mean increased relocalization to granules. **h**, Coefficients and AU content for $k$-mer features. **i**, Coefficients for codon features.

cytoskeleton components localized in the periphery of membranes during the UPR.

## Discussion

In this work, we present an integrative multiomics subcellular localization framework composed of two different methods (LoRNA and dLOPIT) to generate the first cell-wide analysis of RNA and protein subcellular localization in membranous (nucleolus, ER and mitochondria) and membraneless (cytosol, nucleolus and cytosolic granules) compartments. By precisely reconstructing RNA localization, our transformative approach allows the quantitative determination of the complete subcellular distribution of each RNA. We combined LoRNA and dLOPIT to characterize the dynamic transcriptome and proteome subcellular redistribution upon UPR. This established the RNA features driving relocalization to granules, identified those transcripts that are targeted to the periphery of the organelles during the integrated stress response, and revealed the role of eIF3d in maintaining cytoskeleton function upon UPR.

When interpreting LoRNA proportions, it is important to consider that the RNA marker profiles that are used to estimate localization proportions do not represent spatial coordinates in the cell, but rather the

typical profile for RNAs that predominantly reside in that localization. For example, the distribution of the cytosolic RNA markers includes nascent RNA copies that are also nuclear localized. As such, proportions are with respect to RNAs that are paradigmatic representatives of localization, and do not represent absolute localizations. Another important consideration is that, while RNA nuclear export is a tightly regulated process, small proteins diffuse freely. Therefore, when interpreting cell fractionation experiments it is important to consider that small nuclear proteins with weak interactions can diffuse out from the nucleus[60]. Fixing protein–protein interactions could overcome this limitation; however, cross-linking agents such as formaldehyde may limit organelle separation. Development of reagents to efficiently stabilize protein–protein interactions without affecting organelle integrity and lipid content should overcome this limitation. Here we identified nucleoplasm protein markers reflecting this phenomenon to ensure nucleoplasm proteins are correctly classified to the nucleus. Furthermore, in cases where gradients are of interest, such as across the cytosol of a polarized cell, targeted methods such as FISH or IF are required as any high-throughput method involving cell lysis will necessarily result in a loss of subcellular coordinates. When RNA compartments not amenable to biochemical fractionation are of interest (for example,

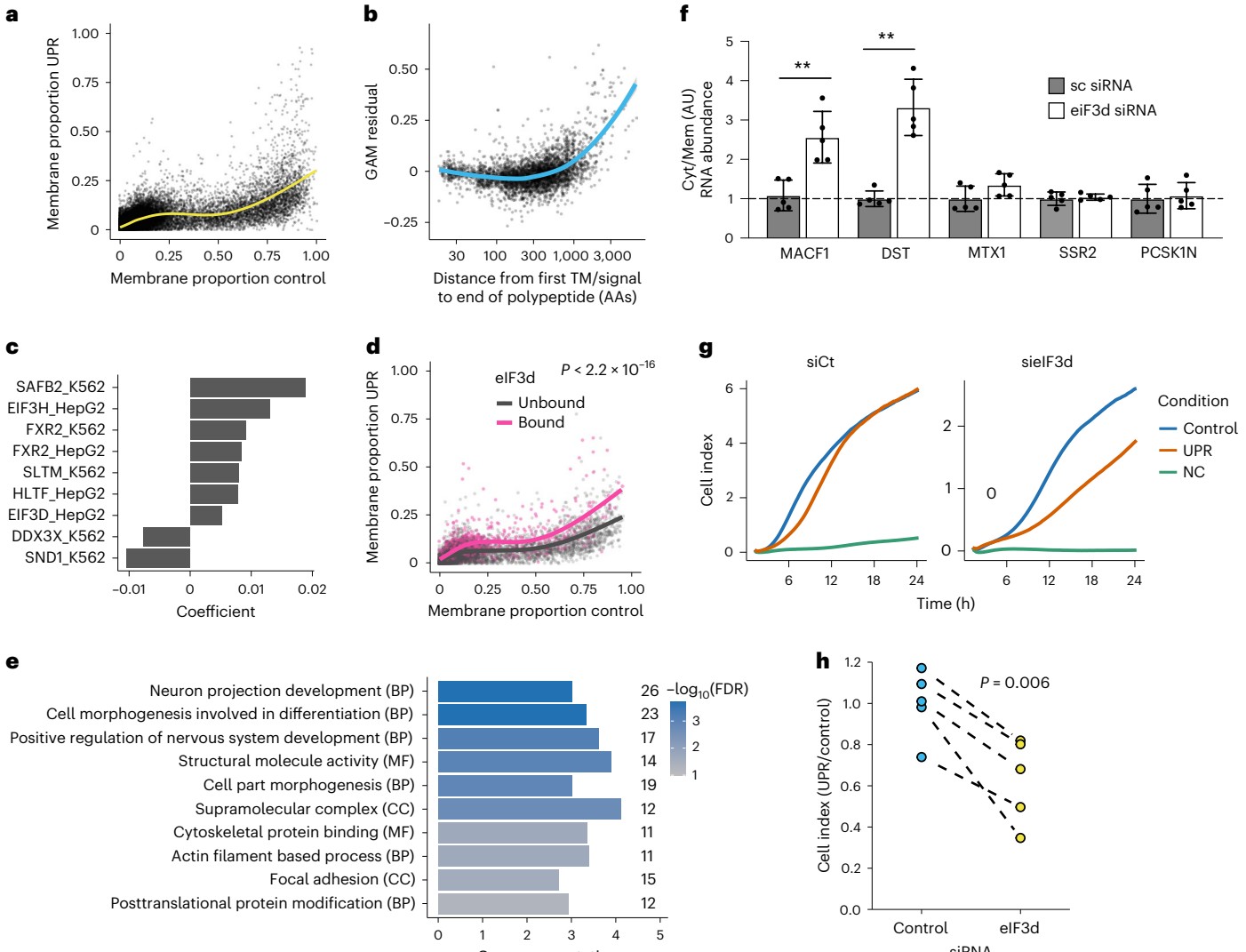

**Fig. 6 | Analysis of the RNAs retaining membrane association under UPR.**
**a**, Membrane proportions in control and UPR. Yellow line indicates fit from GAM with cubic regression spline. **b**, Relationship between residual from GAM and the distance between the first signal or TM domain and stop codon. Blue line indicates smoothed fit by LOESS local regression. AA, amino acid. **c**, Coefficients for RBPs that are predictors of GAM residuals for RNAs that do not encode a signal peptide/TM domain. RBP eCLIP cell line is indicated in the feature name. *P* value from analysis of variance comparing GAM ± stratification by eIF3d binding. **d**, Membrane proportions as per **a**, for RNAs bound by eIF3d according to eCLIP or Subunit-seq. **e**, GO terms enriched in membrane RNAs bound by eIF3d, relative to all membrane RNAs. BP, biological process. CC, cellular compartment. MF, molecular function. **f**, Cytosolic (Cyt) and membrane

(Mem)-associated RNA extraction using digitonin under UPR induction. MTX1, SSR2 and PSK1N are non-eIF3d ER-associated controls. All RNAs are normalized to the MT-ND6 housekeeping transcript. Changes in RNA localization upon eIF3d siRNA are relative to the RNA localization in the scrambled siRNA condition. Error bars represent standard deviation (*n* = 5 independent cytosol and membrane RNA extractions per condition). **P = 0.0079 (two-tailed Mann–Whitney *U* test). **g**, Representative U-2 OS cell migration time series for eIF3d knockdown and siRNA control (siCt) cells in control and UPR conditions. Cell index denotes impedance measured from migrating cells. **h**, Quantification of cell migration from **g** at 24 h post UPR induction in eIF3d knockdown and siRNA control cells (*n* = 5 independent experiments). *P* value derived from a two-tailed paired *t*-test.

the nucleopore), alternative methods like APEX-seq may be applicable. However, it is worth noting that multiple APEX-based experiments need to be combined to determine RNA localization at a cell-wide level, making this approach incompatible with the proportional estimation of RNA localization and especially challenging to apply in dynamic systems or to study multiple biological systems. Nonetheless, APEX and LoRNA can function as complementary approaches to interrogate molecular subcellular distribution at different resolutions.

One of our most striking findings regarding RNA relocalization after UPR activation is that membrane-localized mRNAs are recruited more efficiently to granules than cytosolic mRNAs, while lncRNAs do not relocalize to granules. We show that the two features most associated with mRNA recruitment to granules, length and AU content,

do not affect lncRNA relocalization to granules. This suggests that RNA recruitment to granules may depend on more than RNA length and ribosome occupancy as previously stated[23] or that there is a mechanism to exclude lncRNAs from granules. Notably, membrane-localized mRNA recruitment to granules conflicts with transcriptomic studies that have claimed ER-targeted RNA are depleted from SGs[23]. However, targeted methods to purify SGs typically focus on specific engineered bait proteins and require multiple purification steps, while LoRNA recovers granules regardless of their specific composition, based on their distinct density sedimentation profile, which may explain the apparent discrepancy between SG transcriptomics and LoRNA. Since our results suggest ER-RNA relocalization to granules during stress is widespread, we speculate that novel subtypes of stress-induced granules may be

the ultimate destination for membrane-associated RNAs upon UPR activation, opening new avenues for future research. Furthermore, we observed previously uncharacterized predictors of RNA relocalization to granules. Notably, the relocalization of RNA to granules is correlated with frequencies for three NAU/C codons that are decoded by queuosine modification of the G in the wobble position of the cognate tRNA, with opposite effects for NAU and NAC codons. Considering the emerging role that tRNA modifications play in translation[61,62], it would be interesting to explore if a tRNA-specific co-translational mechanism could regulate the recruitment of RNAs to SGs.

Using dLOPIT under UPR induction, we revealed the relocalization of specific proteins, many of which have been previously implicated in the formation of P-bodies and SGs. We also observed relocalization of ZNF622 to the ER upon UPR and that RNAs containing ZNF622 binding sites relocalize more readily to granules. Given that ZNF622 has been recently shown to inhibit ribosome subunit joining[52], and may be involved in the ER-associated degradation machinery or ER quality control following viral infection[54], ZNF622 relocalization to the ER may be required to inhibit ER-localized translation and drive ER-targeted RNAs to granules upon UPR activation.

The precise measurement of RNA proportionality afforded by LoRNA allowed characterization of the relocalization of RNA away from membranes in unprecedented detail. Intriguingly, we found that, despite the general loss of mRNA from the ER upon induction of the UPR[63], many RNAs are partially retained in association with organelles, or even migrate to the ER. Notably, we found that many of these mRNAs encode for proteins involved in cytoskeletal remodeling. Furthermore, eIF3d binding correlates with membrane localization and decreasing its expression impairs cell migration, a process highly dependent on cytoskeletal remodeling. Actin cytoskeleton remodeling plays a key role to restore $Ca^{2+}$ homeostasis in the ER upon UPR induction through promotion of ER–plasma membrane contact[59]. Separately, eIF3d has been found to maintain RNA translation upon stress[56]. Altogether, we speculate that the cytoskeleton remodeling required to overcome UPR is eIF3d dependent and that it involves the in situ translation of cytoskeletal mRNAs.

In summary, our framework, combining LoRNA and dLOPIT, provides a quantitative system-wide determination of RNA and protein relocalization. This has allowed us to evaluate the transcriptome and proteome dynamics during UPR induction with unprecedented resolution. The resulting data have invaluable potential for future studies characterizing the functions of specific RNAs and proteins. To facilitate this, we have generated a user-friendly graphical interface to explore our data available at refs. 25,26. Notably, the LoRNA method allows for the unbiased cell-wide determination of RNA compartmentalization. This will support a paradigm shift from the study of RNA localization through relative enrichments between two localizations, towards a cell-wide analysis of RNA proportional distribution. We anticipate others will build upon LoRNA. For example, coupling it with metabolic labeling such as 4-thiouridine, and direct RNA-seq methods, would enable the precise temporal interrogation of the role of RNA modifications in the regulation of RNA localization. The simultaneous characterization of proteome and transcriptome relocalization represents a transformative approach to study how the cell coordinately responds to physiological signals, stress conditions, exogenous cues or infectious pathogens. Importantly, this approach can contribute to the translation of molecular biology observations to medical benefits by fostering new studies into the role of RNA and protein localization dynamics in cell homeostasis and disease.

## Online content

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

## Methods

### Cell culture and UPR induction

U-2 OS cells were obtained from the American Type Culture Collection, maintained in McCoy's A5 medium (Gibco-BRL) supplemented with 10% of fetal bovine serum (Gibco-BRL), at 37 °C and 5% CO$_2$, and regularly tested for mycoplasma contamination with negative results. UPR was induced by directly adding 250 nM of TG (UPR) or equivalent volume of dimethyl sulfoxide (DMSO, control) to cells at 90% confluency. Cells were incubated with TG or DMSO for 1 h at 37 °C unless specified otherwise.

### Density-based cell fractionation

Seven milliliters discontinuous density gradients of 15%, 20% and 25% iodixanol (OptiPrep, STEMCELL Technologies), 0.25 M sucrose, 75 mM KCl, 5 mM MgCl$_2$, 50 nM CaCl$_2$, 10 mM HEPES pH 7.4 and ethylenediaminetetraacetic acid (EDTA)-free protease inhibitor were prepared in polyallomer optiSeal ultracentrifuge tubes (11.2 ml capacity; Beckman Coulter), and gradients were allowed to diffuse 1 h at 20 °C. Partially diffused gradients were stored at 4 °C for 1 h while cells were prepared for fractionation. Cells were cultured in 500 mm$^2$ plates until 90% confluence, using a single plate per replica per condition. Cells were treated with TG or DMSO for 1 h. After treatment, cells were washed twice with phosphate-buffered saline (PBS) and detached using EDTA-free trypsin (Thermo Scientific) for 10 min. Trypsin was quenched by adding equal volumes of medium (supplemented with DMSO or TG). Detached cells were transferred to a 50-ml tube and spun down 10 min at 250$g$. Cell pellets were washed twice with ice-cold PBS and resuspended in 900 µl of lysis buffer (0.25 M sucrose, 75 mM KCl, 5 mM MgCl$_2$, 50 nM CaCl$_2$, 10 mM HEPES pH 7.4 and EDTA-free protease inhibitor) and lysed with a ball-bearing homogenizer (Isobiotec) on ice. Fifty microliters of lysate was stored at −80 °C as total cell lysate. Cell lysate iodixanol and ion concentration was adjusted by adding 1.5 ml of 50% iodixanol solution (in 75 mM KCl, 5 mM MgCl$_2$, 50 nM CaCl$_2$, 10 mM HEPES pH 7.4 and EDTA-free protease inhibitor) to a 1-ml cell lysate, and underlaid in the previously prepared density gradient with a 2.5-ml syringe and a wide-bore blunt-end needle (Sigma-Aldrich). Finally, a 40% iodixanol (in 75 mM KCl, 5 mM MgCl$_2$, 50 nM CaCl$_2$, 10 mM HEPES pH 7.4 and EDTA-free protease inhibitor) cushion was underlaid until the tube was filled. Density gradients were centrifuged in a NVT65 fixed-angle near-vertical ultracentrifuge rotor (Beckman Coulter) in a Optima L-80 XP ultracentrifuge (Beckman Coulter) for 16 h at 100,000$g$ and collected using an auto Densi-Flow peristaltic pump fraction collector with a meniscus-tracking probe (Labconco) to obtain 20 fractions of 500 µl each. The refractive index (RI) of each fraction is measured with a hand-held refractometer (Reichert), and the iodixanol concentration was calculated as iodixanol% = (RI/0.83) − 10.111, and fraction density calculated by $d = m/V$. The iodixanol concentration per fraction was adjusted to 30% in a volume of 600 µl. All fractions were frozen and dried by sublimation using vacuum centrifuge with cold trap (Labconco, Refrigerated CentriVap concentrator). Dried pellets were solubilized in 1 ml of TRIzol (Thermo Scientific) and stored at −80 °C.

### Differential sedimentation speed-based cell fractionation

Cells were cultured in 500-mm$^2$ plates until 90% confluence, using a single plate per replicate. Five replicates were performed for the differential sedimentation speed based cell fractionation experiment. Cells were washed twice with PBS and detached using trypsin/EDTA (0.05%) (Thermo Scientific) for 5 min. Trypsin was quenched with an equal volume of medium. Detached cells were transferred to a 50-ml Falcon tube and spun down for 5 min at 200$g$ at 4 °C. The pellets were washed twice with ice-cold PBS and resuspended in 1 ml of lysis buffer (0.25 M sucrose, 75 mM KCl, 5 mM MgCl$_2$, 50 nM CaCl$_2$, 10 mM HEPES pH 7.4 and EDTA-free protease inhibitor) and homogenized on ice using ball-bearing homogenizer (Isobiotec). A total lysate sample of 75 µl was obtained and stored at −80 °C. The remaining sample was fractionated into five consecutive fractions at centrifugation speeds (100$g$, 500$g$, 2,000$g$ and 5,000$g$) using the supernatant of every centrifugation as starting material for the next, with an Eppendorf Centrifuge 5424R. The supernatant of the last centrifugation was retained as the final fraction.

### RNA and protein sample precipitation

RNA and protein were obtained from TRIzol solubilized fractions by adding 200 µl of chloroform and phase partitioning the sample for 15 min at 12,000$g$ at 4 °C. RNA was purified by collecting and transferring the TRIzol/chloroform upper aqueous phase to a new tube and the RNA precipitated with 750 µl of isopropanol (Sigma-Aldrich) for 10 min at 16,000$g$. RNA pellets were washed twice with 70% ethanol and solubilized in 200 µl of RNAse-free water (Thermo Scientific). RNA samples were treated with DNAse in RNeasy columns (Qiagen) according to manufacturer's instructions using the RNase-free DNAse Set kit (Qiagen). RNA concentration was measured with a DS-11 UV spectrophotometer (Denovix). RNA samples were pooled as indicated in Supplementary Table 4. Protein was purified by precipitating the TRIzol/chloroform lower organic phase (and interface) using 9:1 v:v of methanol:sample. Samples were solubilized in 1% sodium dodecyl sulfate (Thermo Fisher Scientific) 100 mM tetraethylammonium bicarbonate (Sigma-Aldrich) using a Bioruptor sonicating bath (Diagenode). Protein concentration was measured with a Pierce BCA protein concentration assay kit (Thermo Fisher Scientific) on a spectrophotometer plate reader (Molecular Devices, SpectroMax M2).

### RNA-seq

RNA-seq libraries for the density fractionation were generated using 1 µg of RNA as starting material and depleting ribosomal RNA using Ribocop V3 (Lexogen). Post ribosomal RNA depletion, RNA content was measured with a Bioanalyzer pico kit (Agilent). Then 1% RNA spike-in RNA variants (SIRVs) were spiked-in (Lexogen), and RNA-seq libraries were generated and amplified using the CORALL RNA-seq kit (Lexogen) according to the manufacturer's instructions. All CORALL-generated libraries were sequenced in parallel on four Novaseq S4 lanes (Illumina). RNA-seq libraries for the differential sedimentation speed based cell fractionation experiment were performed using QuantSeq 3' mRNA-Seq kit (Lexogen) according to manufacturer's instructions. A total of 400 ng of total RNA and 0.1% RNA SIRVs (Lexogen) were used for library preparation of each fraction. All libraries were balanced, multiplexed and pooled, and run on two single-end Novaseq SP lanes (Illumina).

### Proteomic sample preparation

Samples were resuspended in 100 µl of 100 mM tetraethylammonium bicarbonate (Sigma-Aldrich), reduced with 10 mM dithiothreitol (DTT, Sigma-Aldrich) at room temperature (RT) for 60 min and alkylated with 40 mM iodoacetamide (Sigma-Aldrich) at RT in the dark for 60 min. Samples were digested overnight at 37 °C with 1 µg of trypsin (Promega). Subsequently, 1 µg of modified trypsin (Promega) was added, and the samples were incubated for 3–4 h at 37 °C. Samples were then acidified with trifluoroacetic acid (TFA) (0.5% (v/v) final concentration (Sigma-Aldrich) and centrifuged at 21,000$g$ for 10 min. The supernatant was immediately desalted.

For peptide clean-up and quantification, 200 µl of Poros Oligo R3 (Thermo Fisher Scientific) resin slurry (approximately 150–200 µl resin) was packed into Pierce centrifuge columns (Thermo Fisher Scientific) and equilibrated with 0.1% TFA. Samples were loaded, washed twice with 200 µl 0.1% TFA and eluted with 300 µl 70% acetonitrile (ACN) (adapted from ref. [64]). From each elution, 10 µl was taken for Qubit protein assay (Thermo Fisher Scientific) quantification and the remaining sample was retained for MS.

Tandem mass tag (TMT)-10plex or TMTpro-16plex (Thermo Fisher Scientific) labeling from desalted peptides was performed according to the manufacturer's protocol. Equal amounts of desalted peptides

were labeled immediately after being quantified with Qubit protein assay (Thermo Fisher Scientific). Multiplexed TMT samples were then fractionated using high-pH reverse phase chromatography. In detail, the TMT-labeled peptide samples were resuspended in 100 µl of 20 mM ammonium formate pH 10 (Buffer A). The total volume of each sample was injected onto an Acquity UPLC BEH C18 column (2.1-mm inner diameter × 150 mm; 1.7-µm particle size) on an Acquity UPLC System with a diode array detector (Waters), and the peptides were eluted from the column using a linear gradient of 4–60% (v/v) ACN in 20 mM ammonium formate pH 10 over 50 min and at a 0.244 ml min$^{-1}$ flow rate (with a total run time of 75 min). The gradient was set up as follows: 0 min−95% Buffer A−5% Buffer B (20 mM ammonium formate pH 10 + 80% (v/v) ACN), 10 min−95% Buffer A−5% Buffer B, 60 min−25% Buffer A−75% Buffer B, 62 min−0% Buffer A−100% Buffer B, 67.5 min−0% Buffer A−100% Buffer B, 67.6 min−95% Buffer A−5% Buffer B. Approximately 40−50 1-min fractions, representing peak peptide elution, were collected starting from initial peptides elution, and were reduced to dryness by vacuum centrifugation shortly thereafter. For downstream MS analysis, the fractions were concatenated into 20 samples by combining pairs of fractions that eluted at different time points during the gradient.

Each sample was analyzed in an Orbitrap Eclipse mass spectrometer (Thermo Fisher Scientific). Mass spectra were acquired in positive ion mode applying data acquisition using synchronous precursor selection MS3 (SPS-MS3) acquisition mode[65] triggered using Real-time Search (RTS) against human protein sequences from UniProt/Swiss-Prot. Carbamidomethylation of cysteine and TMT-6plex (total proteome samples) or TMTpro-16plex tagging (subcellular fractionated samples) of lysine and peptide N-terminus were set as static modifications, with oxidation of methionine as a variable modification. Scoring thresholds were set as follows: Xcorr = 1.4, dCn = 0.1 and Precursor ppm = 10.

### MS spectra processing and peptide and protein identification

Raw data were viewed in Xcalibur v.2.1 (Thermo Fisher Scientific), and data processing was performed using Proteome Discoverer v2.3 (Thermo Fisher Scientific). The raw files were submitted to a database search using Proteome Discoverer with SequestHF and MS Amanda[66] algorithms against the *Homo sapiens* database downloaded in June 2020 from UniProt/Swiss-Prot. Common contaminant proteins (several types of human keratin, bovine serum albumin (BSA) and porcine trypsin) from the common Repository of Adventitious Proteins (cRAP) v1.0 (48 sequences, adapted from the Global Proteome Machine repository[67]) were added to the database. The spectra identification was performed with the following parameters: MS accuracy, 10 ppm.; MS/MS accuracy of 0.5 Da; up to two missed cleavage sites allowed; carbamidomethylation of cysteine and TMT-6plex (total proteome samples) or TMTpro-16plex tagging (subcellular fractionated samples) of lysine and peptide N-terminus as a fixed modification; and oxidation of methionine and deamidation of asparagine and glutamine as variable modifications. Percolator was used for false discovery rate (FDR) estimation and only rank 1 peptide identifications of high confidence (FDR <1%) were accepted. TMT reporter values were assessed through Proteome Discoverer v2.3 using the Most Confident Centroid method for peak integration and integration tolerance of 20 ppm. Reporter ion intensities were adjusted to correct for the isotopic impurities of the different TMT reagents (following manufacturer specifications). Sample labels for each TMT tag are presented in Supplementary Table 4.

### Spatial proteomics

Previous marker proteins defined for U-2 OS were annotated in ref. 6. Twelve of 733 markers were deemed outliers on the basis of manual curation of their profile and consideration of localization assigned in Human Protein Atlas (HPA)[68] and GO. To ensure accurate assignment of nuclear proteins, nuclear markers were annotated de novo, utilizing the COMPARTMENTS database[69]. COMPARTMENTS localizations were mapped to the localizations defined by the standard LOPIT marker sets. Proteins with a score of 5 for the 'Nucleus' and no score over 2 for any other localization were denoted as exclusively nuclear. The profiles for these nuclear exclusive proteins were split into three groups, which were separated on the basis of the following thresholds on the row-sum normalized abundances: group 1, over 0.3 in pooled fraction 4; group 2, over 0.2 in pooled fraction 5; group 3, over 0.4 in pooled fraction 8. A GO enrichment analysis, using a hypergeometric test, was then used to determine the enriched functionalities, relative to the background of all quantified proteins. This indicated that the first group of nuclear proteins were highly enriched in nucleolus, chromatin and ribosome biogenesis proteins, whereas the other two were enriched in nucleoplasm proteins but no more specific GO terms. Furthermore, proteins annotated as nuclear membrane or nuclear lamina in GO were more closely associated with group 1. Thus, group 1 was denoted as 'Nucleus' and the other groups were referred to as 'Nucleoplasm-1' and 'Nucleoplasm-2'. Finally, the 'Proteosome' marker set was expanded to define a set of markers for 'Protein complexes' by including proteins annotated in GO as part of the multiple aminoacyl-tRNA synthetase (MARS) complex ('GO:0017101'), COP9 signalosome ('GO:0008180') or eIF3 complex ('GO:0005852'), with 5/42 of the additional protein complex markers excluded as outliers. SVM classification in basal conditions was performed as described in ref. 31, with hyperparameters selected by grid search and 50 iterations.

BANDLE[39] v1.0 was used to identify differentially localized proteins between the control condition and UPR. Differential localization analysis was performed on each replicate separately, with 10,000 Markov chain Monte Carlo iterations, 5,000 burn-in iterations, 4 chains and 1/20 thinning. Off-diagonal values for the matrix of Dirichlet priors were set at 0.01, with default values used for the penalized complexity priors. Markov chain Monte Carlo chains were inspected for convergence on the basis of the reported number of localization outliers, as suggested in ref. 39, and all chains were found to converge. Localizations in each condition were determined by setting the following threshold: bandle localization probability × (1 − bandle outlier probability) > 0.95. Where this threshold was not met, the localization was deemed 'undefined'. Proteins were deemed differentially localized if they were never assigned to the same localization across the two conditions and at least 2/3 replicates had BANDLE differential localization probabilities over the following thresholds to define three levels of confidence were for relocalization: ≥0.99, 'highly confident'; ≥0.95, 'confident'; ≥0.85, 'candidate'.

### Comparison of dLOPIT with other methods

Subcellular location data (The Cell Atlas) was downloaded from HPA (version 22.0) and compared with dLOPIT localization classifications obtained with BANDLE and hyperLOPIT and LOPIT-DC classifications[6]. LOPIT and HPA classifications were standardized to eight localizations that were interrogated with both approaches; cytosol, ER, Golgi apparatus, lysosome, mitochondria, nucleus, peroxisome and plasma membrane. For the HPA classifications, we considered cytosol to be cytoplasmic bodies, and nucleus to be nuclear bodies, nuclear membrane, nuclear speckles, nucleoli fibrillar center, nucleoli and nucleoplasm. For the LOPIT classifications, we considered nucleus to be nucleoplasm-1, nucleoplasm-2 and chromatin, and cytosol to be protein complex, proteasome, ribosome 40S, ribosome 60S and ribosome. All other classifications not part of the standard localizations were discarded. Proteins not classified by both LOPIT and HPA were excluded.

HPA may classify a single protein with multiple localizations. For each protein, agreement between HPA and LOPIT was deemed to have occurred when the LOPIT classification was within the set of HPA classifications. HPA classifies at increasing levels of confidence, from Approved, to Supported, Enhanced and Uncertain. We separately

considered all HPA classifications meeting a minimal level of confidence from Approved to Enhanced.

### RNA-seq data processing for CORALL RNA-seq samples

RNA-seq fastq processing and quantification was performed using bespoke pipelines built with CGAT-core[70]. Fastq files were demultiplexed using idemux[71] and concatenated into one fastq per sample. To assess polymerase chain reaction duplication rate, UMIs were extracted from the read sequences using UMI-tools v1.1.0 (ref. [72]). Reads were aligned to a concatenation of the hg38 reference genome and the artificial SIRV genome using hisat v2.2.1 (ref. [73]) using default settings. Secondary reads and reads with MAPQ <10 were discarded. Duplicate reads were identified with UMI-tools dedup using default settings. Transcript isoform quantification was performed from the fastqs without deduplication, using Salmon v1.4.0 (ref. [74]) against a concatenation of the ensembl v102 human transcriptome and the artificial SIRV annotations, using default settings.

### Data analysis

Data analyses post RNA and peptide quantification steps above were performed using R v4.0.3 (ref. [75]) and R markdown notebooks[76], making extensive use of the tidyverse v1.3.2R packages[77], ggplot2 v3.4.2, tidyr v1.2.0, dplyr v1.0.8 and MSnbase v2.20.4 (ref. [78]), pRoloc v1.34.0 (ref. [79]) and camprotR v0.0.0.900 (ref. [80]).

### Differential abundance analysis

Gene-level quantifications (transcripts per million; TPM) were parsed using tximport[81]. Differential gene abundance was tested using DESeq2 v1.34.0 (ref. [82]) with default settings, with an FDR threshold of 5% used to identify significant changes in abundance.

### Spatial transcriptomics

Transcript isoform and gene-level quantifications (TPM) were parsed using tximport v1.22.0 (ref. [81]) to generate objects to hold the quantification estimates for a single experiment. To take advantage of the spatial proteomics functionalities available through pRoloc[79], we stored the RNA quantification data in MSnSets. Separate objects were created to hold transcript isoform and gene-level abundance estimates. Transcripts and genes with average TPM <0.5 or TPM of 0 in ≥2/3 samples in a given condition were discarded. Post TPM-filtering, 41,547 transcript isoforms and 14,203 genes were quantified in at least 2 replicates in both conditions.

Localization markers were identified using a combination of a priori markers and semi-supervised clustering by non-negative matrix factorization (NNMF). A priori markers were defined as follows. Nuclear markers were >16-fold nuclear enriched in nuclear/cytoplasm fraction RNA-seq[29] or manually defined known nuclear lncRNAs (XIST, MALAT1, MEG3, DLX6-AS1, PINCR, UCHL1-AS1 and NEAT1). Cytosol markers were significantly enriched in nuclear export signal (NES) APEX-seq[11], or manually defined from known cytosolic lncRNAs (LINCMD1, NORAD, H19, NKILA, SNHG5, DANCR, OIP5-AS1 and SNHG1). ER markers were defined as having enrichment in ER-Ribo-Seq >2^0.5 (ref. [30]) and significant enrichment >8-fold in ER (KDEL) APEX-seq[11] and a predicted signal peptide or TM according to ensembl. Mitochondrial markers were mitochondrially encoded mRNAs. To determine the optimal number of clusters (k) for NNMF, the imputation-based approach was used[83], with the selected k value minimizing the mean squared error for NNMF imputation of randomly added missing values. This gave k = 5 for the basal condition experiment and k = 4 for the UPR experiment. NNMF cluster assignments in basal conditions were then compared with the a priori markers to define a set of NNMF-guided gene-level markers, in which each marker set was associated with the NNMF cluster with greatest overlap and all markers in the NNMF cluster were retained. In addition, the NNMF cluster containing the nuclear markers was further used to define a

nucleolus marker set by intersecting it with the top 30 most abundant small nucleolar RNAs according to the total RNA-seq. Finally, a novel NNMF cluster was observed in both basal condition and UPR, which contained few a priori markers. The genes assigned to the novel NNMF cluster in both basal and UPR conditions were used to define a novel profile, which was observed to have the greatest relative abundance in between the nucleus and cytosol profile peaks, and was hence denoted as cytosol light. Gene-level markers were used to generate transcript-level markers by taking all the transcript isoforms for each gene-level marker where the ensembl transcript biotype and gene biotype matched. The final marker sets were manually curated to remove 14/135 gene-level markers and 37/202 transcript-level markers that were deemed to be outliers. The higher proportion of transcript-level markers removed reflects the markers having been built at the gene level and extended naively to transcript-level markers by assuming all transcripts with the same biotype as the gene biotype will have the same localization.

### Estimation of localization proportions

RNA content per fraction was estimated using the relative abundance of all SIRV features and human RNA and the proportion of the fraction used for RNA-seq library preparation. Relative RNA-seq quantifications in each fraction were adjusted with respect to the RNA content per fraction and abundance estimates row-sum normalized across the eight fractions per replicate. Non-negative least squares regression was used to estimate localization proportions by separately modeling the profile of each transcript/gene as a non-negative linear combination of the average profile for the markers of each localization. Since most of the ER markers relocalized upon UPR and the mitochondrial markers represent 100% membrane localization in both conditions (no RNA copies in the cytosol), mitochondrial markers were used to estimate the membrane proportion. The proportion estimates were bootstrapped 100 times by resampling the markers, with resampling. Proportion estimates for a given transcript/gene were discarded where the model did not account for at least 90% of the variance or the absolute value or the intercept was greater than 0.05. The mean proportions across replicates were estimated for transcripts/genes with estimates from at least 2/3 replicates. In total, proportions for 27,368 transcript isoforms and 13,274 genes were retained.

Reads were frequently observed to bridge annotations between lncRNAs and neighboring protein coding, which resulted in mis-estimation of lncRNA proportions. To avoid this, we excluded all lncRNAs with a protein-coding gene within 15 kb downstream or 30 kb upstream.

### Comparison of LoRNA with other methods

The comparison of LoRNA proportions with other methods was performed using gold standards for nuclear and ER localization, namely nuclear/cytoplasm fractionation RNA-seq and ER Ribo-seq. Comparisons were performed with gene-level data, since transcript-level data were not available for all experiments. Gene-level quantification (TPM) from fractionation polyA + RNA-seq[29] was obtained from www.encodeproject.org. Mean nuclear/cytoplasm ratios were calculated across the cell lines and genes with ratios quantified in at least 2/9 cell lines were retained. ER Ribo-seq data[30] was converted from UNIPROTKB identifiers to ensembl gene IDs using the R biomaRt package. APEX-seq[11] data were obtained from Gene Expression Omnibus GSE116008. APEX-RIP[9] data were obtained from Gene Expression Omnibus GSE106493. Cell fractionation RNA-sequencing (CeFra-seq)[14] data were taken and nuclear/cytosol and membrane/cytosol ratios calculated. Relative enrichments from LoRNA were calculated from proportions as for example LocalizationX/(1 − LocalizationX), with nuclear proportions being the sum of nucleus and nucleolus proportions. All enrichments were log2-transformed and Pearson correlation coefficients against gold-standard ratios were calculated.

The classification of ribosome-associated lncRNAs was obtained from ref. [36]. lncRNAs with multiple transcript IDs were excluded. Ribosome profiling data were obtained from ref. [84] and the coding sequence (CDS) length obtained from ensembl. Ribosome density was then calculated as RNaseI_CDS/CDS length. P-body RNAs were obtained from[48]. RNAs with over 8-fold higher abundance in the sorted fraction and Benjamini–Hochberg[85] adjusted *P* value less than 0.01 were deemed P-body RNAs.

MERFISH data for ER and nuclear enrichment was obtained from ref. [12]. These were presented at the level of RNA transcripts and were therefore compared with transcript-level LoRNA proportions.

### Defining signal peptide and TM features

Signal peptide and TM domain annotations were obtained from ensembl v102 and the distance between the start of the first signal peptide or TM domain and the stop codon was determined. For gene-level analyses, the minimum distance across all transcript isoforms was used. A conservative annotation of presence/absence of either signal peptide and/or TM domain was obtained by taking the union of the ensembl annotations with UNIPROT annotations.

### eCLIP binding data

eCLIP data in narrowPeak (BED6 + 4) format was obtained from ENCODE using ENCODExplorer v2.16.0. Genomic coordinates were converted to transcriptomic coordinates using the mapToTranscripts function in GenomicFeatures v1.46.5 R package[86].

### Modelling lncRNA cytosol localization

To model the cytosolic localization of lncRNAs, transcript length, spliced status, eCLIP binding data, *k*-mer frequencies, AU content, RNA modifications, PolyA status, presence in FANTOM5 robust catalog, and abundance were considered. Transcript length and splicing status was obtained from ensembl v102. eCLIP data were obtained as described above. Sixty-four RBPs with >10 lncRNA targets were retained, and binding data were converted to binary 0 = unbound, 1 = bound. *k*-mers ($k = 1–7$) were counted and expressed as frequencies. RNA modifications were obtained from m6A-atlas[87], with modifications with >10 lncRNA targets retained, namely m6A, m5c, m1A and Psi. The FANTOM5 robust catalog and PolyA status were obtained from ref. [88]. PolyA status was converted to a binary 0 = non-polyadenylated, 1 = polyadenylated, with 'undetermined' encoded as 0 and 'bimorphic' encoded as 1. Abundance was calculated as the mean TPM from the total RNA-seq of basal condition samples. Cytosol proportions were converted to a binary feature, where 1 = cytosolic (2/3 cytosol) and 0 = not-cytosolic (<1/3 cytosol). The data were then split 80:20 into training and test data. Elastic net logistic regression was used to model the cytosol variable, with 10-fold cross-validation, using the glmnet v4.1-3R package[89]. By default glmnet scales the dependent variables but returns the coefficients on the original scale. $\alpha$ values were varied between 0 and 1, in steps of 0.1. Cross-validation folds were precomputed to ensure they were the same for each $\alpha$ value. The $\alpha$ value with the minimum mean cross-validation error was identified as 1, for example lasso regression. Lambda was selected to give the most parsimonious model, using the 'one-standard-error' rule[90]. The final model contained 45/21,918 nonzero coefficients. The predictive accuracy of this model was assessed using the hold-out test data and compared with the accuracy of a simpler logistic regression model of just transcript length, polyA status and AU content.

### Modeling UPR-resistant membrane localization

Membrane proportion in UPR was modeled as being dependent upon membrane proportion in control conditions using a GAM with a cubic regression spline with shrinkage, using the mgcv R package. The residual from the GAM was taken to represent the degree of UPR-resistant localization relative to the overall reduction in membrane proportions.

Lasso regression (glmnet R package[89]) was used to model UPR resistance, with RBP binding from eCLIP data (see above) being the dependent variables. Ten-fold cross-validation was used to select the lambda value that minimized the mean cross-validation error, with the 'one-standard-error' rule used to select the most parsimonious model[90]. Transcripts with signal peptides/TM domains were transcripts modeled separately from those without.

### Modelling relocalization to granules

Granule relocalization (UPR granule proportion − control granule proportion) was modeled according to the following features, which were annotated to each mRNA: localization in basal conditions, RNA length, RBP binding, sequence *k*-mers, 5′ AUGs, codon features, uORFs and IRES. One-hot encoding was used for the features describing the localization in basal conditions, where the highest proportion localization was taken to be the single localization for the RNA. Transcript, 5′ UTR, 3′ UTR and coding sequence lengths were obtained from ensembl v102. eCLIP data were obtained as described above. A total of 177 RBPs with >100 mRNA transcript targets were retained, and binding data were converted to binary 0 = unbound, 1 = bound. *k*-mers ($k = 1–6$) were counted and expressed as frequencies. 5′ AUGs were identified with separate features to encode in-frame and out-of-frame AUGs. Codon, dinucleotide and wobble base frequencies were computed from the coding sequence. Codon optimality (MILC[91]) was computed, with separate features for the comparison against all transcripts and just the top 1% most abundant. uORFs were identified from ref. [92], retaining only ORFs with a score >5 and an AUG start codon. Annotated IRES were obtained from IRESbase[93]. uORF and IRES were converted to binary presence/absence features for each transcript. In addition, pairwise interactions between the localization in basal conditions and the transcript, 3′ UTR, 5′ UTR and coding length features and RBP binding features were added as separate features.

Granule relocalization data was split 80:20 into training and test data. Elastic net regression was used to model the granule relocalization, with 10-fold cross-validation, using the glmnet R package[89]. By default glmnet scales the dependent variables but returns the coefficients on the original scale. $\alpha$ values were varied between 0 and 1, in steps of 0.1. Cross-validation folds were precomputed to ensure they were the same for each $\alpha$ value. The $\alpha$ value with the minimum mean cross-validation error was identified as 1, for example, lasso regression. Lambda was selected to give the most parsimonious model, using the 'one-standard-error' rule[90]. The final model contained 74/16,895 nonzero coefficients.

### Data processing and analysis for differential centrifugation-based LoRNA

Data processing and analysis for samples quantified using 3′ Quant Seq was identical to CORALL RNA-seq samples, except for the following: Gene read counts were normalized to counts per million (CPM) using the total number of assigned reads per sample. Genes with average CPM <1 or CPM of 0 in > 20% of the samples were discarded. Markers identified from equilibrium density centrifugation-based LoRNA were annotated and proportions calculated as indicated above, except that proportions were estimated separately for mitochondria and ER and summed to give membrane proportions, and 'cytosol-light' proportions were not estimated. The mean proportions across replicates were estimated for transcripts/genes with estimates from at least 3/5 replicates.

### Assessing the technical bias for RNA length and sedimentation

Gene-level CeFra-seq quantification data (TPM) were downloaded using the ENCODExplorer v2.16.0 R package. We computed the gene length as the mean length of all transcript isoforms included in the GENCODE basic gene set and with Transcript Support Level 1 in ensembl. Mean abundances per fraction were computed for the cytosol and

membrane fractions, from which ratios were then computed compared with the total RNA-seq samples. High-confidence cytosolic and membrane localized RNAs were obtained from LoRNA, setting a threshold of proportion >80%. Pearson product-moment correlation coefficients were computed for gene length versus cytosol/total for cytosolic RNAs and membrane/total for membrane RNAs. The same procedure was used for the differential sedimentation-based RNA localization approach estimates, with supernatant/total correlated with gene length for cytosolic RNAs.

## GO enrichment analyses

All GO enrichment analyses were performed using the goseq v1.46.0 R package[94]. For the analysis of GO terms enriched in eIF3d-bound membrane RNAs, the membrane proportion in basal conditions was included as the bias factor and enrichment effect sizes accounting for biasing factors were calculated using the probability weight functions obtained with goseq using the estimate_go_overrep function in camprotR. For the analysis of GO terms enriched in the membrane RNAs not encoding a signal peptide or TM domain, no bias factor was included and a hypergeometric test was instead used. For both analyses, $P$ values were adjusted for multiple testing using the Benjamini–Hochberg procedure[85] and GO terms with adjusted $P$ value >0.05 (5% FDR) or accounting for fewer than 10% of the foreground genes were excluded. Redundant GO terms were removed using the remove_redunant_go function in camprotR R package.

## Single molecule FISH probe design and synthesis

Subcellular RNA localization was assessed by using an adaptation of the single-molecule inexpensive FISH protocol[95]. Z (CTTATAGGGCATG-GATGCTAGAAGCTGG) and Y (AATGCATGTCGACGAGGTCCGAGTGTAA) FLAP DNA handles, labeled at the 5' and 3' ends with Atto488, were purchased from Sigma (high-performance liquid chromatography (HPLC) purified) and resuspended to a concentration of 100 µM in nuclease-free water. For each target RNA, 30–48 DNA probes of 20 nucleotides (nt) were designed with a minimum spacing length of 2 nt and a guanine–cytosine content of 40–65%. Each gene-specific sequence was flanked by a 28-nt sequence complementary to either a Z- or Y-FLAP sequence. The resulting 48-nt probes were purchased from Sigma (standard desalt purification, 100 µM in nuclease-free water). Fluorescently labeled gene-specific probes were then generated as follows: 200 pmol of an equimolar mixture of all gene-specific oligos for each gene were mixed with 250 pmol of the appropriate FLAP oligo in 1× NEBuffer 3 (New England Biolabs, B7003), then incubated in a Thermocycler (Bio-Rad) for 3 min at 85 °C, 3 min at 65 °C, and 5 min at 25 °C (lid 99 °C).

## smiFISH/IF and IF

For smiFISH/IF experiments, $8 \times 10^4$ U 2-OS cells were seeded in each well of a 12-well plate, on top of no. 1.5 glass coverslips previously washed in 1 M HCl. The following day, cells were treated with either 250 nM TG or the corresponding volume of DMSO, rinsed three times with PBS (with $MgCl_2$ and $CaCl_2$, Sigma D8662), then fixed for 10 min in 3% methanol-free paraformaldehyde (Alfa Aesar, 43368) in PBS at RT. The fixative was then quenched in 100 mM glycine (Sigma) in PBS for 10 min at RT, and samples were then washed twice in PBS for 10 min and permeabilized in 70% EtOH at 4 °C for at least 1 h. Samples were then prepared for hybridization by washing in 10% formamide (Sigma), 1 U µl⁻¹ RNasin Plus (Promega) in 2× saline-sodium citrate (SSC) buffer for 10 min. From this step onwards, samples were protected from direct light. Hybridization was performed by incubating coverslips with 100–250 nM probes diluted in hybridization buffer (2× SSC buffer, 10% dextran sulfate (Sigma), 10% formamide, 2 mM ribonucleoside vanadyl complexes (Sigma), 200 µg ml⁻¹ bovine serum albumin (Roche), 1 mg ml⁻¹ *Escherichia coli* tRNA (Roche), 1 U µl⁻¹ RNasin Plus (Promega)), for 3 h at 37 °C in a humid chamber. Coverslips were then transferred

to a clean 12-well plate and washed twice in 10% formamide, 1 U µl⁻¹ RNasin Plus, 2× SSC buffer, for 10 min at 37 °C. Further washes were then performed (three times in 2× SSC with no incubation, twice in PBS for 10 min), before incubation with blocking buffer (3% nuclease-free bovine serum albumin (Sigma) in PBS) for 30 min at RT. Samples were then incubated with primary antibody diluted 1:500 in blocking buffer, for 2 h at RT. They were then washed three times in PBS for 10 min, incubated with a secondary antibody diluted 1:2,000 in blocking buffer, for 1 h at RT. After three additional washes in PBS, the nucleus was stained by incubation with 4′,6-diamidino-2-phentylindole (Sigma, 200 ng ml⁻¹ in PBS) for 1 min at RT. Coverslips were then washed twice in PBS for 5 min before being mounted onto glass microscope slides with a drop of ProLong Glass Antifade Mountant (Invitrogen).

For consistency, IF experiments were performed following the smiFISH/IF procedure with the omission of the FISH hybridization step and the washes in 10% formamide, 2× SSC.

Digitonin cytosol extraction for smiFISH/IF experiment was performed following the protocol described in ref. 96. Briefly, cells grown on glass coverslips were washed twice in warm CHO buffer (115 mM KAc, 25 mM HEPES pH 7.4, 2.5 mM $MgCl_2$, 2 mM egtazic acid and 150 mM sucrose). The coverslips were then placed (cell-side down) on a droplet of 0.025% digitonin in CHO buffer, on top of a metal plate warmed to 40 °C, for 20 s. Cells were then immediately blocked in 4% paraformaldehyde/PBS for 15 min at RT, then extensively washed with PBS and immersed in ice-cold MetOH for 30 min. After additional washes in PBS and two washes in 10% formamide/SSC buffer, hybridization with FISH probes and IF were performed as described above.

## Confocal microscopy

Images were acquired on a Zeiss Axio Observer.Z1 LSM 980 microscope equipped with Airyscan 2, using ZEN Blue software (version 3.3), in Airyscan super-resolution imaging mode. A C-Plan-Apochromat 63×/1.4 numerical aperture oil objective was used with Zeiss Immersol 518F (23 °C) immersion oil. Twenty $Z$-slices per image were acquired at an interval of 0.13 µm.

## Image processing and quantification

Airyscan processing with standard parameters was applied to each image in Zen Blue software. Further processing and analysis was performed using FiJi software[97] (version 2.3.051).

## siRNA transfection

Transfections were performed in detached cells post trypsinization to ensure maximum exposure. Per condition, 10 nM of each small interfering RNA (siRNA) was transfected using Lipofectamine RNAiMAX (ThermoFisher) according to the manufacturer's instructions. Knockdown experiments were performed using ON-TARGETplus SMARTpool siRNAs (Horizon): control siRNA #1 #D-001810-01-05; eIF3d siRNA #L-017556-00-0005; eIF3j siRNA #L-019532-00-0005. All subsequent experiments were performed 48 h post transfection.

## Cell migration assay

Cell migration assays were performed using the xCELLigence RTCA DP instrument (ACEA Biosciences) according to the manufacturer's instructions. Cells were treated with TG for 1 h, collected by trypsinization and washed in PBS to remove all traces of serum. A total of 30,000 cells were seeded in the upper chamber of a 16-well migration plate (CIM-16 plate) in 100 µl of medium containing 0.1% serum. Cells migrate to a lower chamber containing 160 µl of medium supplemented with 10% serum. As cells migrate across the microelectrodes into the lower chamber they generate impedance measurements (cell index), which enables label-free quantification of cell migration for 24 h. Cell indexes for Tg-treated samples were expressed as fold changes relative to the DMSO control sample. Significance testing between eIF3d and control siRNA knockdowns was performed with a paired Student's $t$-test.

## Digitonin cytosolic RNA extraction

Digitonin extraction protocol was adapted from ref. 98. Cytosolic and membrane-associated RNA fractionation experiments were performed in 80% confluent 12 multiwell plates. Cells were washed twice in ice-cold PBS and incubated for 5 min in 200 µl of digitonin extraction buffer (0.03% of digitonin (ThermoFisher) 5 mM KCl, 5 mM MgCl$_2$, 50 nM CaCl$_2$, 10 mM HEPES pH 7.4 and EDTA-free protease inhibitor). The supernatant was transferred to a new tube, and the cells were incubated for 5 min with 200 µl of digitonin wash buffer (0.004% of digitonin (ThermoFisher) 5 mM KCl, 5 mM MgCl$_2$, 50 nM CaCl$_2$, 10 mM HEPES pH 7.4 and EDTA-free protease inhibitor). The second supernatant was transferred to the same tube as the first one. This combined fraction is enriched in cytosol transcripts. The plates were then scraped in 400 µl of digitonin extraction buffer. This second fraction is enriched in membrane-associated RNAs. Cytosolic and membrane-associated RNA fractions were frozen and dried by sublimation using a vacuum centrifuge with cold trap. RNA was extracted and DNAse treated using the RNeasy kit according to the manufacturer's instructions. A total of 200 µg of RNA were reverse transcribed using Moloney murine leukemia virus reverse transcriptase and random decamers (Ambion). Quantitative polymerase chain reaction amplification reaction was performed using the LightCycler 480 SYBR Green I Master Mix (Roche Diagnostics) and the following set of primers: MACF1 Fw 5′-TAGAGATGACTGCTGTGGC-3′ and MACF1 Rv 5′-TGTCTTGTAACCTCATCTTCGA-3′, DST Fw 5′-ATTGGTACAGAGGGTTGCA-3′ and DST Rv 5′-CGTCCTTTGCTGTACACAG-3′, TMX1 Fw 5′-ACGGACGAGAACTGGAGAGA-3′ and TMX1 Rv-5′-ATTTTGACAAGCAGGGCACC-3′, SSR2 Fw 5′-GTTTGGGATGCCAACGATGAG-3′ and SSR2 Rv 5′-CTCCACGGCGTATCTGTTCA-3′, PSK1N Fw 5′–3′ and PSK1N Rv 5′–3′. Quantitative expression data were normalized to the mitochondrial transcript MT-ND6 using the MT-ND6 Fw 5′-GGGTTGAGGTCTTGGTGAGT-3′ and MT-ND6 Rv 5′-ACCAATCCTACCTCCATCGC-3′ primers. TMX1, SSR2, PSK1N and MT-ND6 primers were obtained from ref. 9.

## Cell death assay

All flow cytometry data were acquired using a BD LSRFortessa (BD Bioscience) and analyzed with BD FACSDiva software (version 9.0.1). A total of 10,000 counts were acquired for each experimental condition. Cells were collected in Annexin binding buffer (BD Biosciences #556454), and cell death was determined after incubation with Annexin-V-FITC (ThermoFisher, BMS500FI-100) and Draq7 (Abcam, ab109202).

## Liquid chromatography–tandem MS acquisition

TMT-labeled samples were analyzed in an Orbitrap Eclipse coupled to a nanoLC Dionex Ultimate 3000 UHPLC (Thermo Fisher Scientific). Peptides were trapped on a 100 µm × 2 cm, C18, 5 µm, 100 trapping column (Acclaim PepMap 100) in µl-pickup injection mode at 15 µl min⁻¹ flow rate for 3 min. Samples were then loaded on a rapid separation liquid chromatography, 75 µm × 50 cm nanoViper C18 3 µm 100 column (Acclaim, PepMap) at 50 °C retrofitted to an EASY-Spray source with a flow rate of 300 nl min⁻¹. Analytical chromatography was performed over 120 min (buffer A, HPLC H$_2$O, 0.1% formic acid; buffer B, 100% ACN, 0.1% formic acid; 0–3 min: at 2% buffer B, 3–105 min: linear gradient 2% to 40% buffer B, 105–105.3 min: 40% to 90% buffer B, 105.3–110 min: at 90% buffer B, 110–110.3 min: 90% to 3% buffer B, 100.3–120 min: at 3% buffer B). Each MS1 scan was performed in the Orbitrap analyzer (mass range of $m/z$ 400–1,500, resolution of 120,000). Precursors with charge between 2 and 6 and intensity above 5,000 were selected for collision-induced dissociation MS2 fragmentation, with normalized automated gain control (AGC) target of 200% and maximum accumulation time of 50 ms. Mass filtering was performed by the quadrupole with 0.7 $m/z$ transmission window, followed by collision-induced dissociation fragmentation in the linear ion trap with 30% normalized collision energy. Selected fragmented ions were dynamically excluded for 60 s. RTS was used to trigger SPS-MS3 acquisition. RTS used Human UniProt/Swiss-Prot database, carbamidomethylation of cysteine and TMT-6plex (total proteome samples) or TMTpro-16plex tagging (subcellular fractionated samples) of lysine and peptide N-terminus as static modification, oxidation of methionine as variable modification, as scoring thresholds Xcorr = 1.4, dCn = 0.1, Precursor PPM = 10, one missed cleavage and maximum search time of 35 ms. SPS was applied to co-select ten fragment ions for higher-energy collisional dissociation-MS3 analysis. SPS ions were all selected within the 400–1,500 $m/z$ range and were set to preclude selection of the precursor ion and TMT or TMTpro ion series. Normalized AGC targets and maximum accumulation times were set to 200% and 120 ms. Co-selected precursors for SPS-MS3 underwent HCD fragmentation with 55% normalized collision energy and were analyzed in the Orbitrap with a nominal resolution of 50,000. The number of SPS-MS3 spectra acquired between full scans was restricted to a duty cycle of 3 s.

## Proteomics data processing

Peptide-spectrum match (PSM)-level quantifications were filtered to conservatively remove peptides from common contaminants. Alongside the cRAP proteins, further potential contaminants were identified by considering all proteins that shared an observed peptide with a cRAP protein as further contaminants. PSMs without a unique master protein assigned or more than 20% missing values were excluded. Remaining missing values were imputed using knn imputed ($k$ = 10), with sum normalization before imputation and de-normalization post imputation, to ensure nearest neighbors shared similar abundance profiles over the fractions, rather than similar average abundance. We then identified and removed outlier PSMs that had a median euclidean distance over 0.2 from all other PSMs for the same master protein. In doing so, PSMs with higher co-isolation and lower average signal to noise were selectively removed where these low-quality PSMs disagreed with other PSMs. Remaining PSMs were then median center normalized. Protein-level abundances were estimated by summing PSM-level abundances, for proteins with at least two PSMs. Protein abundances were row-sum normalized such that the total abundance across all fractions from a given replicate equaled one.

## Polysome profiling

The 10%–50% (w/v) sucrose gradients were prepared in gradient buffer (100 mM NaCl, 5 mM MgCl$_2$, 15 mM Tris–HCl pH 7.5, 1 mM DTT and 0.1 mg ml⁻¹ cycloheximide). Cells were washed in PBS–cyclohexamide (100 g ml⁻¹) and scraped into lysis buffer (100 mM NaCl, 5 mM MgCl$_2$, 15 mM Tris–HCl pH 7.5, 1 mM DTT, 0.2 M sucrose, 0.1 mg ml⁻¹ cycloheximide, 0.5% IGEPAL, and 5 µl RNasin per 1 ml). Lysates were incubated on ice for 3 min, and cells were pelleted by centrifugation at 1,300$g$ for 5 min. The supernatant was layered on top of a gradient and centrifuged at 247,767$g$ (acceleration 9, deceleration 6) for 2 h at 4 °C using a Beckman Coulter ultracentrifuge. Polysome profiles were obtained by measuring absorbance at 254 nm using a UA-6 UV–Vis detector (Presearch).

## Reporter gene expression analysis

DCP2–GFP tagged mammalian cell expression plasmid was directly acquired from Adgene (pT7–EGFP–C1–HsDCP2, ref. 25031). G3BP1–GFP plasmid was kindly provided by Dr. Dee Scadden (University of Cambridge). Plasmids were purified using EasyPep Mini MS Sample Prep Kit (Qiagen). One plate of 500 mm² of 90% confluent U-2 OS cells were transfected per plasmid, per condition using 100 µg of plasmid with Lipofectamine 3000 Transfection Reagent (Thermo Scientific) according to manufacturer instructions. Two days post-transfection, cells were treated with DMSO or TG as indicated above and lysed and fractionated as specified in the 'Density-based cell fractionation' section. Gradient fractions were imaged on a Zeiss Axio Observer.Z1 LSM

980 microscope and fluorescence intensity quantified with ImageJ 1.8.0_172 for Mac OS X.

## Reporting summary

Further information on research design is available in the Nature Portfolio Reporting Summary linked to this article.

## Data availability

The MS proteomics data have been deposited to the ProteomeXchange Consortium via the PRIDE[99] partner repository with the dataset identifier PXD030456. The RNA-seq data have been deposited in ENA with study accession PRJEB49479. The processed data including the data underlying the Shiny app are available as serialized R objects in RDS format from https://github.com/CambridgeCentreForProteomics/LoRNA_UPR (v1.0 and archived with Zenodo, https://doi.org/10.5281/zenodo.8375646). RNA annotations were obtained from ensembl v102 using biomaRt R package. FANTOM5 data were downloaded from https://www.nature.com/articles/s41587-021-00936-1. eCLIP and CeFra-seq data were downloaded from https://www.encodeproject.org. The hyperLOPIT and LOPIT-DC data were obtained using R package pRolocdata (object names: hyperLOPITU2OS2018, lopitdcU2OS2018). The following additional publicly available datasets were used. In all cases the data used are available from https://github.com/CambridgeCentreForProteomics/LoRNA_UPR v1.0 in the directory 1_external. HPA data were downloaded from https://www.proteinatlas.org/about/download on 3 May 2023. MERFISH data were downloaded from https://www.pnas.org/doi/10.1073/pnas.1912459116#data-availability. Ribosome-associated lncRNA classification was obtained from https://www.ncbi.nlm.nih.gov/pmc/articles/PMC5975437. ER Ribo-seq data were obtained from https://www.science.org/doi/10.1126/science.1257521. P-body enriched RNA data were obtained from https://doi.org/10.1016/j.molcel.2017.09.003. APEX-seq data were obtained from https://www.ncbi.nlm.nih.gov/geo/query/acc.cgi?acc=GSE116008. APEX-RIP data were obtained from https://www.ncbi.nlm.nih.gov/geo/query/acc.cgi?acc=GSE106493. IRES data were downloaded from IRESbase (http://reprod.njmu.edu.cn/cgi-bin/iresbase/download.php#human). uORF data were downloaded from https://doi.org/10.1093/nar/gky188. GO term annotations were downloaded from ensembl v102 using biomaRt R package. Data pertaining to the effect of eIF3D Knockdown on translation were downloaded from https://doi.org/10.1016/j.molcel.2020.06.003 (Supplementary Information; Supplementary Data 1). 5′ cap eIF3d binding data was obtained from https://doi.org/10.1126/science.abb0993. Signal peptide and TM annotations were downloaded from Uniprot (https://www.uniprot.org/) using the search string '(annotation:(type:signal) OR annotation:(type:transmem)) AND organism: 'Homo sapiens (Human) [9606]' on 6 September 2021. Data pertaining to SG enrichment upon arsenite treatment were downloaded from https://doi.org/10.1016/j.molcel.2017.10.015. Data pertaining to TIS-granules were downloaded from https://doi.org/10.1101/2022.11.04.515216. RNA modification data were downloaded from m6A Atlas (http://180.208.58.19/m6A-Atlas/download.html) on 2 November 2022.

## Code availability

Data analysis code is available from https://github.com/CambridgeCentreForProteomics/LoRNA_UPR (v1.0), archived with Zenodo (https://doi.org/10.5281/zenodo.8375646).

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

## Acknowledgements

We thank C. De Matos Ferraz Franco of the MRC-Toxicology Proteomics Facility for collection of all MS data and D. Scadden for kindly providing the G3BP1–GFP expression plasmid, B. Fisher for kindly sharing equipment and M. Marti-Solano for assistance with the manuscript writing. We also thank the Advanced Light Imaging facility at the MRC Toxicology Unit and the Light Imaging Facility at the Human Technopole for their support in the acquisition of light microscopy data. E.V., T.S., R.M.L.Q., M.P. and M.E. are supported by Wellcome Trust, grant numbers 110170/Z/15/Z and 110071/Z/15/Z awarded to A.E.W. and K.S.L.; R.F.H, V.D. and M.M. are supported by the Medical Research Council, grant number 5TR00. L.M.B. is supported by EU Horizon 2020 programme INFRAIA project EPIC-XS (project 823839). O.M.C. was supported by a Wellcome Trust Mathematical Genomics and Medicine studentship funded by the Cambridge School of Clinical Medicine and by the Todd-Bird Junior Research Fellowship from New College, Oxford.

## Author contributions

Conceptualization, E.V., T.S., A.E.W. and K.S.L.; methodology, E.V., T.S., M.P. and M.E.; investigation, E.V., T.S., M.P., M.E., R.M.L.Q. and R.F.H.; writing—original draft, E.V., T.S. and M.P.; resources, L.M.B.; data curation, T.S.; writing—review and editing, E.V., T.S., M.P., M.E., R.M.L.Q., R.F.H., O.M.C., M.M., V.D., A.E.W. and K.S.L.; visualization, E.V., T.S. and M.P.; supervision, A.E.W. and K.S.L.; project administration, E.V., T.S., A.E.W. and K.S.L.; funding acquisition, A.E.W. and K.S.L.

## Competing interests

The authors declare no competing interests.

## Additional information

**Correspondence and requests for materials** should be addressed to Anne E. Willis or Kathryn S. Lilley.

# Reporting Summary

## Statistics

For all statistical analyses, confirm that the following items are present in the figure legend, table legend, main text, or Methods section.

| n/a | Confirmed | |
|---|---|---|
| ☐ | ☒ | The exact sample size (*n*) for each experimental group/condition, given as a discrete number and unit of measurement |
| ☐ | ☒ | A statement on whether measurements were taken from distinct samples or whether the same sample was measured repeatedly |
| ☐ | ☒ | The statistical test(s) used AND whether they are one- or two-sided<br>*Only common tests should be described solely by name; describe more complex techniques in the Methods section.* |
| ☐ | ☒ | A description of all covariates tested |
| ☐ | ☒ | A description of any assumptions or corrections, such as tests of normality and adjustment for multiple comparisons |
| ☐ | ☒ | A full description of the statistical parameters including central tendency (e.g. means) or other basic estimates (e.g. regression coefficient) AND variation (e.g. standard deviation) or associated estimates of uncertainty (e.g. confidence intervals) |
| ☐ | ☒ | For null hypothesis testing, the test statistic (e.g. $F$, $t$, $r$) with confidence intervals, effect sizes, degrees of freedom and $P$ value noted<br>*Give P values as exact values whenever suitable.* |
| ☐ | ☒ | For Bayesian analysis, information on the choice of priors and Markov chain Monte Carlo settings |
| ☐ | ☒ | For hierarchical and complex designs, identification of the appropriate level for tests and full reporting of outcomes |
| ☐ | ☒ | Estimates of effect sizes (e.g. Cohen's *d*, Pearson's *r*), indicating how they were calculated |

*Our web collection on statistics for biologists contains articles on many of the points above.*

## Software and code

Policy information about availability of computer code

| | |
|---|---|
| Data collection | Confocal microscopy: Images were acquired using ZEN Blue software v3.3 and processed using FiJi software v2.3.051. |
| Data analysis | Proteomics data processing: Raw data were viewed in Xcalibur v.2.1. Peptide spectrum matching was performed with Proteome Discoverer v2.3 with SequestHF and MS Amanda algorithm<br><br>Transcriptomics data processing: Fastq files were demultiplexed using idemux (https://github.com/Lexogen-Tools/idemux). UMIs were extracted using UMI-tools v1.1.0. Reads were aligned using hisat v2.2.1. Quantification was performed using Salmon v1.4.0.<br><br>Data analyses was performed using R v4.0.3 and R markdown notebooks, using multiple packages, predominantly ggplot2 v3.4.2, tidyr v1.2.0, dplyr v1.0.8, MSnbase v2.20.4, pRoloc v1.34.0, camprotR v0.0.0.900 (https://github.com/CambridgeCentreForProteomics/camprotR), tximport v1.22.0, DESeq2 v1.34.0, BANDLE v1.0, ENCODExplorer v2.16.0, GenomicFeatures v1.46.5, glmnet v4.1-3 and goseq v1.46.0.<br><br>Data analysis code is available from https://github.com/CambridgeCentreForProteomics/LoRNA_UPR (v1.0) and archived with zenodo, DOI: 10.5281/zenodo.8375646 . |

For manuscripts utilizing custom algorithms or software that are central to the research but not yet described in published literature, software must be made available to editors and reviewers. We strongly encourage code deposition in a community repository (e.g. GitHub). See the Nature Portfolio guidelines for submitting code & software for further information.

# Data

Policy information about availability of data

All manuscripts must include a data availability statement. This statement should provide the following information, where applicable:
- Accession codes, unique identifiers, or web links for publicly available datasets
- A description of any restrictions on data availability
- For clinical datasets or third party data, please ensure that the statement adheres to our policy

The mass spectrometry proteomics data generated in this study have been deposited to the ProteomeXchange Consortium via the PRIDE partner repository with the dataset identifier PXD030456.

The RNA-Seq data generated in this study have been deposited in ENA with study accession PRJEB49479.

RNA annotations were obtained from ensembl v102 using biomaRt R package

FANTOM5 data was downloaded from https://www.nature.com/articles/s41587-021-00936-1

eCLIP and CeFra-Seq data were downloaded from https://www.encodeproject.org

The hyperLOPIT and LOPIT-DC data were obtained using R package pRolocdata (object names: hyperLOPITU2OS2018, lopitdcU2OS2018)

The following additional publicly available datasets were used. In all cases the data used is available from https://github.com/CambridgeCentreForProteomics/LoRNA_UPR (v1.0; archived with zenodo, DOI: 10.5281/zenodo.8375646) in the directory 1_external

Human Protein Atlas data was downloaded from https://www.proteinatlas.org/about/download on 03 MAY 2023

MERFISH data was downloaded from https://www.pnas.org/doi/10.1073/pnas.1912459116#data-availability

Ribosome-associated lncRNAs classification was obtained from https://www.ncbi.nlm.nih.gov/pmc/articles/PMC5975437

ER Riboseq data was obtained from https://www.science.org/doi/10.1126/science.1257521

P-body enriched RNA data was obtained from DOI: https://doi.org/10.1016/j.molcel.2017.09.003

APEX-Seq data was obtained from https://www.ncbi.nlm.nih.gov/geo/query/acc.cgi?acc=GSE116008

APEX-RIP data was obtained from https://www.ncbi.nlm.nih.gov/geo/query/acc.cgi?acc=GSE106493

IRES data was downloaded from IRESbase (http://reprod.njmu.edu.cn/cgi-bin/iresbase/download.php#human)

uORF data was downloaded from https://doi.org/10.1093/nar/gky188

GO term annotations were downloaded from ensembl v102 using biomaRt R package

Data pertaining to the effect of eIF3D Knockdown on translation was downloaded from https://doi.org/10.1016/j.molcel.2020.06.003 (Supplemental Information; Data S1)

5' cap eIF3d binding data was obtained from DOI: 10.1126/science.abb0993

Signal peptide and transmembrane domain annotations were downloaded from Uniprot (https://www.uniprot.org/) using the search string '(annotation: (type:signal) OR annotation:(type:transmem)) AND organism:"Homo sapiens (Human) [9606]"' on 06 SEP 21

Data pertaining to stress granule enrichment upon arsenite treatment were downloaded from https://doi.org/10.1016/j.molcel.2017.10.015

Data pertaining to TIS-granules were downloaded from DOI: 10.1101/2022.11.04.515216

RNA modification data was downloaded from m6A Atlas (http://180.208.58.19/m6A-Atlas/download.html) on 02 Nov 2022

# Human research participants

Policy information about studies involving human research participants and Sex and Gender in Research.

| | |
|---|---|
| Reporting on sex and gender | NA |
| Population characteristics | NA |
| Recruitment | NA |

| Ethics oversight | NA |
|---|---|

Note that full information on the approval of the study protocol must also be provided in the manuscript.

# Field-specific reporting

Please select the one below that is the best fit for your research. If you are not sure, read the appropriate sections before making your selection.

☒ Life sciences ☐ Behavioural & social sciences ☐ Ecological, evolutionary & environmental sciences

For a reference copy of the document with all sections, see nature.com/documents/nr-reporting-summary-flat.pdf

# Life sciences study design

All studies must disclose on these points even when the disclosure is negative.

| Sample size | No statistical methods were used to predetermine sample size. Sample sizes were chosen based on experience of the relevant techniques and an assessment of the observed variance. |
|---|---|
| Data exclusions | No data was excluded |
| Replication | Density-based LoRNA: 3 independent experiments were performed. Moreover, an orthogonal sedimentation-based LoRNA was developed which verified the accuracy of the localisation proportions estimates and ensured they were not biased by the fractionation approach. This sedimentation-based LoRNA was applied in 5 independent experiments.<br><br>Cell migration assays: 5 independent experiments were performed which verified the reproducibility of the observed effect of eIF3D knockdown.<br><br>smFISH/IF: 3 independent experiments were performed which verified the reproducibility of the observed relocalisation of RNA upon activation of the UPR. |
| Randomization | Cell culture plates were randomly assigned to experimental conditions. |
| Blinding | Blinding was not required in our experimental design since we did not allocate to groups and data processing and analysis were performed identically for all samples, precluding human bias. |

# Reporting for specific materials, systems and methods

We require information from authors about some types of materials, experimental systems and methods used in many studies. Here, indicate whether each material, system or method listed is relevant to your study. If you are not sure if a list item applies to your research, read the appropriate section before selecting a response.

## Materials & experimental systems

| n/a | Involved in the study |
|---|---|
| ☐ | ☒ Antibodies |
| ☐ | ☒ Eukaryotic cell lines |
| ☒ | ☐ Palaeontology and archaeology |
| ☒ | ☐ Animals and other organisms |
| ☒ | ☐ Clinical data |
| ☒ | ☐ Dual use research of concern |

## Methods

| n/a | Involved in the study |
|---|---|
| ☒ | ☐ ChIP-seq |
| ☒ | ☐ Flow cytometry |
| ☒ | ☐ MRI-based neuroimaging |

## Antibodies

| Antibodies used | Anti-Calnexin Abcam ab22595<br>Anti-ZNF622 Santa-Cruz sc-10098<br>Goat anti-Rabbit IgG Invitrogen, A-21245<br>Goat anti-Mouse IgG Invitrogen, A-11029<br>Anti-b-tubulin CST #2146<br>Anti-eIF3d Proteintech #66024-1-Ig<br>Anti-fibrillarin Cell Signaling C13C3<br>anti-histone H3 Bethyl Laboratories #A300-823A<br>anti-beta actin Abcam ab8227<br>anti-calreticulin Cell Signaling D3E6<br>Anti-COX 1V Cell Signalling 4850 |
|---|---|

| Validation | ab22595 Knockout and WB validated by manufacturer; sc-100980 WB validation by the manufacturer & used in PMID: 26240280; #2146 WB validation by manufacturer & used in PMID: 36869047; #66024-1-Ig WB validation by the manufacturer; C13C3 WB validation by manufacturer & used in: PMID: 34158490; #A300-823A WB validated by the manufacturer; ab8227 WB validation by manufacturer; D3E6 WB validation by the manufacturer; 4850 WB validation by the manufacturer |
|---|---|

## Eukaryotic cell lines

Policy information about cell lines and Sex and Gender in Research

| Cell line source(s) | U-2 OS (U2OS) were obtained from the American Type Culture Collection (ATCC). |
|---|---|
| Authentication | None of the cells were authenticated |
| Mycoplasma contamination | All cells were regularly tested for mycoplasma contamination with negative results |
| Commonly misidentified lines (See ICLAC register) | No commonly misidentified cell lines were used in this study |

