## [Peer Review File · Nature Methods]

Peer Review Information

Manuscript Title: System-wide analysis of RNA and protein subcellular localisation dynamics

Corresponding author name(s): Kathryn Lilley

Editorial Notes: n/a

Reviewer Comments & Decisions:

Decision Letter, initial version:

Dear Kathryn,

Your Article, "A system-wide quantitative map of RNA and protein subcellular localisation dynamics", has now been seen by 3 reviewers. As you will see from their comments below, although the reviewers find your work of considerable potential interest, they have raised a number of concerns. We are interested in the possibility of publishing your paper in Nature Methods, but would like to consider your response to these concerns before we reach a final decision on publication.

We therefore invite you to revise your manuscript to address these concerns. In particular, we think it would be important to further validate the approach as suggested by Reviewer #1, along with all the other reviewers comments. Additionally, we also think that instead of using a new acronym it might be better to describe it as an approach/workflow rather than a novel method.

* include a point-by-point response to the reviewers and to any editorial suggestions

* please underline/highlight any additions to the text or areas with other significant changes to facilitate review of the revised manuscript

* address the points listed described below to conform to our open science requirements

* ensure it complies with our general format requirements as set out in our guide to authors at www.nature.com/naturemethods

* resubmit all the necessary files electronically by using the link below to access your home page

[Redacted] This URL links to your confidential home page and associated information about manuscripts you may have submitted, or that you are reviewing for us. If you wish to forward this email to co-authors, please delete the link to your homepage.

We hope to receive your revised paper within 10 weeks. If you cannot send it within this time, please let us know. In this event, we will still be happy to reconsider your paper at a later date so long as nothing similar has been accepted for publication at Nature Methods or published elsewhere.

OPEN SCIENCE REQUIREMENTS

REPORTING SUMMARY AND EDITORIAL POLICY CHECKLISTS

IMAGE INTEGRITY

DATA AVAILABILITY

All novel DNA and RNA sequencing data, protein sequences, genetic polymorphisms, linked genotype and phenotype data, gene expression data, macromolecular structures, and proteomics data must be deposited in a publicly accessible database, and accession codes and associated hyperlinks must be provided in the "Data Availability" section.

To further increase transparency, we encourage you to provide, in tabular form, the data underlying the graphical representations used in your figures. This is in addition to our data-deposition policy for specific types of experiments and large datasets. For readers, the source data will be made accessible directly from the figure legend. Spreadsheets can be submitted in .xls, .xlsx or .csv formats. Only one (1)

file per figure is permitted: thus if there is a multi-paneled figure the source data for each panel should be clearly labeled in the csv/Excel file; alternately the data for a figure can be included in multiple, clearly labeled sheets in an Excel file. File sizes of up to 30 MB are permitted. When submitting source data files with your manuscript please select the Source Data file type and use the Title field in the File Description tab to indicate which figure the source data pertains to.

Please include a “Data availability” subsection in the Online Methods. This section should inform readers about the availability of the data used to support the conclusions of your study, including accession codes to public repositories, references to source data that may be published alongside the paper, unique identifiers such as URLs to data repository entries, or data set DOIs, and any other statement about data availability. At a minimum, you should include the following statement: “The data that support the findings of this study are available from the corresponding author upon request”, describing which data is available upon request and mentioning any restrictions on availability. If DOIs are provided, please include these in the Reference list (authors, title, publisher (repository name), identifier, year). For more guidance on how to write this section please see: <http://www.nature.com/authors/policies/data/data-availability-statements-data-citations.pdf>

MATERIALS AVAILABILITY

SUPPLEMENTARY PROTOCOL

To help facilitate reproducibility and uptake of your method, we ask you to prepare a step-by-step Supplementary Protocol for the method described in this paper. We [encourage authors to share their step-by-step experimental protocols](https://www.nature.com/nature-research/editorial-policies/reporting-standards#protocols) on a protocol sharing platform of their choice and report the protocol DOI in the reference list. Nature Portfolio's Protocol Exchange is a free-to-use and open resource for protocols; protocols deposited in Protocol Exchange are citable and can be linked from the published article. More details can found at <https://www.nature.com/nature-research/editorial-policies/reporting-standards#protocols>

href="https://www.nature.com/protocolexchange/about"
target="new">www.nature.com/protocolexchange/about.

ORCID

Sincerely,
Arunima

Arunima Singh, Ph.D.
Senior Editor
Nature Methods

Reviewers' Comments:

Reviewer #1:

Remarks to the Author:

Villanueva and co-authors report on the use of density fractionation to isolate biomolecules including RNA and proteins from different subcellular components including organelles (nucleus, ER and mitochondria) and membraneless compartments (cytosol, nucleolus and cytosolic granules). RNA profiling was conducted using the method named LoRNA (Localisation of RNA) and proteins were analyzed using dLOPIT (density-based Localisation of Proteins by Isotope Tagging) via mass spec. The integrated analysis is named SuMM (Subcellular Multiomics Mapping), which is said to be a new

framework to simultaneously interrogate the dynamics of the transcriptome and proteome at subcellular resolution.

Although these methods do not appear to be completely unexpected or totally novel at first glimpse, somehow the findings are very interesting. Specifically, they identified localization dynamics of 31839 transcripts and 5314 proteins during a cellular process called the unfolded protein response (UPR). They observed that while generally mRNAs predominantly localize in the cytosol, RNAs encoding proteins with signal peptides and/or transmembrane domains (TMs) were much more prominently membrane localized, consistently with their co-translational targeting to the ER. Longer 3'UTRs in membrane mRNAs were found to be the greatest positive predictor of relocalization to granules. There are unexpected associations between RBP binding and relocalization to granules, including ZNF622, which is involved in ribosome subunit joining and ribosome stalling, and upregulated upon viral infection and UPR stress. They observed that ER-localized RNAs are recruited to granules to a greater extent than previously thought, RNAs encoding cytoskeletal proteins are retained and targeted to the periphery of organelles during UPR and that this process involves eIF3d. The authors also provided the data through several dedicated open-source resources. Overall, this work is a comprehensive dissection of subcellular compartment-specific RNA and protein localization and dynamics, representing a highly valuable resource for the community.

A concern this reviewer might have had is how pure each density fraction is and how much contamination from different subcellular organelles or compartments. It should be possible to detect or image what constitutes each fraction, which should be used as an important quality assessment process.

The data are very interesting and highly valuable, but may not be appropriate to be called a “map” since this work did not generate a 2D or 3D map of biomolecules in a cell, but relative distribution in different subcellular compartments.

The statement that “these approaches are restricted to the interrogation of a single subcellular compartment per experiment and thus can not provide a cell-wide view of RNA localization” is not really true. Mapping near-transcriptome wide RNA distribution to subcellular location can be done with techniques like MERFISH or seqFISH. It is too much to ask the authors to perform MERFISH or seqFISH to validate their work, but it might be a good idea to choose the same cell lines that have been imaged by MERFISH or seqFISH to do LoRNA, and then compare the results to the published high-plex smFISH data.

Another option is to compare RNA distribution in nucleus vs whole cell using public dataset from the same type of cells characterized by snRNA-seq vs scRNA-seq.

For subcellular protein mapping, a publication from Emma Lundberg (Science 2019, DOI: 10.1126/science.aal3321) is a perfect example. Some authors of this paper are also co-authors in the

Science 2019 paper. Would be possible to perform dLOPIT on the same cell line as in the Science 2019 paper, and then compare or benchmark against the published data?

Since this manuscript discussed the dynamics of RNA localization, is it possible to track RNAs dynamics using either metabolic labeling or computational methods that can track RNA biogenesis in single cells to compare.

Reviewer #2:

Remarks to the Author:

Villanueva et al. report a method for simultaneous fractionation of RNAs and proteins, which allows them to map the location of these molecules and the changes of their locations upon a perturbation. As a whole, I found their results and interpretations supported by the data.

Below are specific comments that can help the authors improve their paper:

– The authors state that transcriptome wide analysis of RNA localization has been a long standing challenge. Yet, I recall multiple papers that reported such analysis about a decade ago. Below are links to a couple of Cell papers reporting transcriptome localization analysis:

<https://doi.org/10.1016/j.cell.2012.06.041>

<https://doi.org/10.1016/j.cell.2012.05.043>

– The novelty of the methodological advances was not very clear to me. It is clear that the authors report the first simultaneous analysis of RNA and protein localization, but it is less clear what are the significant methodological innovations that had to be introduced. I read “We developed a novel fractionation approach that separates the major subcellular localisations while avoiding the RNA length-dependent sedimentation bias. (Fig. 2c & Supplementary Fig. 2b,c,d, Supplementary table 1, 2).”, but did not understand the associated significant methodological innovation.

– While the simultaneous analysis of RNA and protein localization is compellingly demonstrated, the new biological analysis and interpretations that it enables seem less compelling.

– The criteria used to infer RNA localization to granules seem reasonable and consistent with RNA localization to granules though I am not convinced that they prove localization to granules.

– The authors seem to have collected and analyzed 20 off-line fractions but report data for only about 5314 proteins. Why is the proteome coverage not higher ?

– The biological results of the analysis are well described, though my enthusiasm for them is reduced by the use of a non physiological condition, Thapsigargin treatment, which triggers UPR and other cellular responses.

Reviewer #3:

Remarks to the Author:

In this manuscript, the authors present an integrated approach where they combine localization of RNAs to a density-based localization approach. The novelty of this method is that the authors study the subcellular structures by their density and quantify RNA abundance profiles across the density fractions with proteomic measurements made in parallel. They also apply this to study the redistribution of RNA and protein during the activation of UPR. The authors present some interesting findings related to localization of lncRNAs, discovery of membrane-localized RNAs lacking signal peptide or TM domains and recruitment of ER-localized RNAs to granules. I feel that the manuscript is acceptable for publication after the following issues are addressed:

1. The dLOPIT method is a modification to a previously published method (Localization of organelle proteins by isotope tagging (LOPIT)) by the group in 2004 (Mol Cell Proteomics. 2004.3(11):1128-34) but seems to be highlighted here as a completely new method.
2. The discussion about lncRNAs is not clear. I believe that the authors should have data that allows them to correlate their localization findings with several other features:
 - i) Are they able to discover lncRNAs that have been shown to encode proteins based on their data?
 - ii) Can they correlate the proteomic data with translation of the subset of lncRNAs that encode proteins?
3. Overall, the manuscript is hard to read. It would be great if the authors could simplify the manuscript to make it more accessible to a wide audience.
4. The authors do not discuss any data pertaining to mitochondria in the text. Is this because of the overlap with ER? I believe mitochondria would serve as a nice control because of mitochondrial transcription, RNA processing and translation within the organelle and it would be nice to know if the authors found expected correlations in their data related to mitochondria.

5. I am not convinced that this approach deserves another acronym: Subcellular Multiomics Mapping (SuMM). Another reason is that a lot of conclusions are based on correlations and not direct observations/measurements although the term 'mapping' implies greater precision.

Author Rebuttal to Initial comments

Reviewer #1:

Remarks to the Author:

Villanueva and co-authors report on the use of density fractionation to isolate biomolecules including RNA and proteins from different subcellular components including organelles (nucleus, ER and mitochondria) and membraneless compartments (cytosol, nucleolus and cytosolic granules). RNA profiling was conducted using the method named LoRNA (Localisation of RNA) and proteins were analyzed using dLOPIT (density-based Localisation of Proteins by Isotope Tagging) via mass spec. The integrated analysis is named SuMM (Subcellular Multiomics Mapping), which is said to be a new framework to simultaneously interrogate the dynamics of the transcriptome and proteome at subcellular resolution.

Although these methods do not appear to be completely unexpected or totally novel at first glimpse, somehow the findings are very interesting. Specifically, they identified localization dynamics of 31839 transcripts and 5314 proteins during a cellular process called the unfolded protein response (UPR). They observed that while generally mRNAs predominantly localize in the cytosol, RNAs encoding proteins with signal peptides and/or transmembrane domains (TMs) were much more prominently membrane localized, consistently with their co-translational targeting to the ER. Longer 3'UTRs in membrane mRNAs were found to be the greatest positive predictor of relocalization to granules. There are unexpected associations between RBP binding and relocalization to granules, including ZNF622, which is involved in ribosome subunit joining and ribosome stalling, and upregulated upon viral infection and UPR stress. They observed that ER-localized RNAs are recruited to granules to a greater extent than previously thought, RNAs encoding cytoskeletal proteins are retained and targeted to the periphery of organelles during UPR and that this process involves eIF3d. The authors also provided the data through several dedicated open-source resources. Overall, this work is a comprehensive dissection of subcellular compartment-specific RNA and protein localization and dynamics, representing a highly valuable resource for the community.

We would like to thank the reviewer for recognising the value of our work for the community.

A concern this reviewer might have had is how pure each density fraction is and how much contamination from different subcellular organelles or compartments. It should be possible to detect or image what constitutes each fraction, which should be used as an important quality assessment process.

LoRNA and LOPIT are not based on obtaining purified organelles/compartments in a given fraction. Instead, they rely on the principle of correlation profiling^{1,2}, whereby the intensity profile of each macromolecule across the fractions is compared with profiles for macromolecules with known localisations. Importantly, the use of correlation profiles is not only more cost and time efficient than obtaining meticulously purified fractions, but allows the interrogation of partially overlapping localisations, not amenable to purification. We have clarified the underlying principle in the results.

The data are very interesting and highly valuable, but may not be appropriate to be called a "map" since this work did not generate a 2D or 3D map of biomolecules in a cell, but relative distribution in different subcellular compartments.

We agree that the use of the term 'map' could be confusing when we are not using it in the context of 'mapping' macromolecules to a specific organelle or compartment. We have amended the text to avoid using 'map' in these instances.

The statement that "these approaches are restricted to the interrogation of a single subcellular compartment per experiment and thus can not provide a cell-wide view of RNA localization" is not really true. Mapping near-transcriptome wide RNA distribution to subcellular location can be done with techniques like MERFISH or seqFISH.

This statement was meant to apply only to *'proximity-dependent biotinylation techniques, such as APEX-RIP and APEX-seq'*. While MERFISH or seqFISH can be used to obtain the spatial coordinates of each detected RNA, this data lacks contextual information about the localisation of the organelles (except the nucleus), limiting the interpretation of subcellular localisation. Indeed, our understanding is that MERFISH and seqFISH were primarily designed to quantify RNA counts at a single cell level, not to determine subcellular localisation. Combining MERFISH and seqFISH with immunolabeling does enable quantification of the relative proximity of an RNA to specific proteins, which can be used to infer subcellular localisation. However, a comprehensive analysis of the transcriptome compartmentalisation with these approaches has not yet been demonstrated. We have clarified the introduction to make the limitations of existing methods clearer.

It is too much to ask the authors to perform MERFISH or seqFISH to validate their work, but it might be a good idea to choose the same cell lines that have been imaged by MERFISH or seqFISH to do LoRNA, and then compare the results to the published high-plex smFISH data. Another option is to compare RNA distribution in nucleus vs whole cell using public dataset from the same type of cells characterized by snRNA-seq vs scRNA-seq.

We agree that comparing our LoRNA data to MERFISH would be useful and we would like to thank the reviewer for this suggestion. As the reviewer suggested, we have compared our LoRNA U-2 OS results to MERFISH results generated in U-2 OS³, where they contextualised MERFISH RNA coordinates with ER and Nucleoplasm marker proteins. We have compared our LoRNA membrane and Nuclear proportions to MERFISH ER vs non-ER cytoplasm and Nucleus vs cytoplasm, respectively. Both comparisons show a strong positive correlation between MERFISH and LoRNA. We have added this as Supplementary Figure 2i and included the new figure panels below.

Comparison between MERFISH and LoRNA proportions, using data from³. Spearman correlations calculated for RNAs which are significantly enriched in the respective localisations according to MERFISH.

For subcellular protein mapping, a publication from Emma Lundberg (Science 2019, DOI: 10.1126/science.aal3321) is a perfect example. Some authors of this paper are also co-authors in the Science 2019 paper. Would be possible to perform dLOPIT on the same cell line as in the Science 2019 paper, and then compare or benchmark against the published data?

We agree that comparing our dLOPIT data to the Cell Atlas data from the Human Protein Atlas⁴ as suggested would be useful and we would like to thank the reviewer for this suggestion. Importantly, the Cell Atlas data includes the same cell line (U-2 OS)⁴ we have used for the subcellular protein mapping using dLOPIT. We have taken the latest Cell Atlas data and compared the subcellular localisation classifications with dLOPIT and benchmarked against two 'gold-standard' published LOPIT methods (also generated in U-2 OS), LOPIT-DC⁵ and hyperLOPIT⁶. We have added this comparison as Supplementary figure 1h and included the figures below. Considering that dLOPIT is part of an integrated framework for the simultaneous localisation of RNA and protein, and that therefore it has to account for the specific constraints that interrogating both macromolecules impose, the agreement with Cell Atlas is remarkably close to that obtained with the gold-standard LOPITs.

Agreement between Human Protein Atlas (HPA) subcellular localisation and different LOPIT classifications from U-2 OS cells. LOPIT and HPA classify proteins to different subcellular niches. To allow comparison, all niches were simplified to the following: Cytosol, ER, Golgi, Lysosome, Mitochondria, Nucleus, Peroxisome, PM. Niches that could not be mapped onto any localisation, e. Mitotic spindle, were discarded. Only Approved classifications were included from HPA. HPA can annotate multiple localisations for a single protein. Agreement required the LOPIT classification to be within the set of HPA localisations.

Since this manuscript discussed the dynamics of RNA localization, is it possible to track RNAs dynamics using either metabolic labeling or computational methods that can track RNA biogenesis in single cells to compare.

The use of metabolic labelling is something we have considered and we have amended the discussion to mention this.

Reviewer #2:

Remarks to the Author:

Villanueva et al. report a method for simultaneous fractionation of RNAs and proteins, which allows them to map the location of these molecules and the changes of their locations upon a perturbation. As a whole, I found their results and interpretations supported by the data.

We are glad that the reviewer finds our results and interpretations are supported.

Below are specific comments that can help the authors improve their paper:

– The authors state that transcriptome wide analysis of RNA localization has been a long standing challenge. Yet, I recall multiple papers that reported such analysis about a decade ago. Below are links to a couple of Cell papers reporting transcriptome localization analysis: <https://doi.org/10.1016/j.cell.2012.06.041>

<https://doi.org/10.1016/j.cell.2012.05.043>

As we noted in the introduction, previous attempts to determine the localisation of RNA in a cell-wide manner have been restricted by technical limitations, including localisation-independent RNA clustering and length-dependent RNA localisation biases. In particular, in supplementary figure 2B, we show that the high *g*-forces previously used create an RNA length-dependent localisation bias with longer RNAs depleted from the supernatant, resulting in the systematic mislocalization of RNAs in a size-dependent manner.

The method proposed in the first paper indicated, Wang et al., 2012, is essentially a precursor to the CeFra-Seq method proposed in⁷, which we re-analysed in supplementary figure 2B. As we show in our re-analysis, the use of high *g*-force centrifugation steps as used in⁸ (100,000*g*) introduces length biases in the sedimentations and so confounds RNA length and localisation. We have cited the Wang et al publication in the introduction when we mention the technical limitations of previous methods.

The second paper⁹, collects three fractions (cytoplasm, nucleoplasm and chromatin). It therefore can't be considered to provide a comprehensive representation of RNA localisation. Furthermore, the quantification is relative between the three fractions and does not enable the estimation of localisation proportions, which is a significant benefit of LoRNA.

We have amended the introduction to clarify that comprehensive analysis of RNA localisation is not possible using existing methods.

– The novelty of the methodological advances was not very clear to me. It is clear that the authors report the first simultaneous analysis of RNA and protein localization, but it is less clear what are the significant methodological innovations that had to be introduced. I read "We developed a novel fractionation approach that separates the major subcellular localisations while avoiding the RNA length-dependent sedimentation bias. (Fig. 2c & Supplementary Fig. 2b,c,d, Supplementary table 1, 2).", but did not understand the associated significant methodological innovation.

To our knowledge, current methods developed to generate a comprehensive representation of RNA localisation in a contextualised manner (i.e. able to localise the transcriptome to different subcellular compartments, and not just to provide out of context spatial coordinates or interrogate a single localisation) use different versions of out-of-equilibrium differential centrifugation approaches to order the cellular content. These approaches are exemplified in ATLAS-seq¹⁰ and CeFra-Seq⁷ methods.

ATLAS-Seq¹⁰ follows the approach applied by² to study the subcellular localisation of proteins and uses out-of-equilibrium density centrifugation (3h at 30,000 rpm) with material loaded at the top of the gradient. The relative abundance of the RNAs along the gradient is then assumed to represent spatial information. While a similar strategy is used to distinguish ribonucleoproteins according to their composition (ribosome profiling), using it to interrogate the subcellular localisation of RNA is not viable. The main problem arises from the extremely different sedimentation coefficient that different molecules have (determined by the shape, hydration state and density of the molecule between others). RNA and protein heterogeneity generates a gradient of sedimentation profiles that is determined by a combination of their physicochemical properties and their subcellular localisation. To tackle this fundamental problem, we loaded the cell lysate in the gradient at the density of the free-RNA (as shown in supplementary Figure 1) and used a longer centrifugation time and higher speed. This ensured the cytosolic RNAs did not have to travel along the gradient and therefore did not contaminate

other subcellular localisations, while allowing the organelles and their RNA cargo to reach their equilibrium density.

CeFra-Seq⁷ also maps RNA localisation based on the different sedimentation coefficient of the cellular content, but following a pelleting strategy using a smaller volume and centrifugation time. Unfortunately, while this approach works for when mapping the proteome localisation (e.g. in LOPIT-DC⁵), RNAs higher molecular mass makes it co-sediment based on its weight within the *g*-forces used. Thus, RNA will co-sediment with subcellular compartments that is not part of or interacting with. To avoid this limitation when validating our results in figure 2, we limited the centrifugation speed to 5000 g, avoiding the co-sedimentation bias (Sup Fig 2b-d).

From the proteomics perspective, hyperLOPIT⁶ and LOPIT-DC⁵, two gold standard methods developed in the Lilley laboratory to determine proteome subcellular localisation, were designed to specifically map the proteome, not the transcriptome. Both methods use EDTA which chelates divalent ions, disturbing fundamental RNA-protein interactions required to maintain RNA localisation. Moreover, both methods use out-of-equilibrium high *g*-forces making them susceptible to the same length-dependent RNA localisation biases described above. Our dLOPIT approach uses a gradient ion composition designed to maintain RNA-protein interactions, which combined with the loading strategy and centrifugation time and speed, allowed us to develop a new LOPIT method that is compatible with the simultaneous characterisation of the transcriptome and proteome.

Finally, we introduced spike in RNAs to allow the relative RNA abundance profiles to be adjusted for the total RNA content per fraction, and thus enable estimation of localisation proportions instead of localisation enrichments.

In summary, the methodological innovations in this manuscript arise from understanding of the specific requirements to optimally determine RNA and protein localisation. These are:

- Use of equilibrium density centrifugation, with material loaded at the density of free RNA
- Employ an ion composition that maintains RNA-protein interactions
- Spike-in RNAs to allow proportional localisation for RNA

– While the simultaneous analysis of RNA and protein localization is compellingly demonstrated, the new biological analysis and interpretations that it enables seem less compelling.

We are glad that the reviewer found our analysis of the RNA and protein localisation compellingly demonstrated. Our results showcase the benefits of simultaneous RNA and protein localisation analysis, with respect to both the interpretation and biological insights that can be obtained.

Here, one of the benefits of simultaneous RNA and protein localisation analysis is that the localisation of one molecule can inform how to interpret the localisation of the other. This is exemplified when we identify the RNA granule profile. With just the RNA data alone, confident assignment of the novel RNA profiles as representing granules would have been challenging. The UPR-induced movement of canonical stress granules proteins towards the precise same discriminating fraction which defined the novel RNA profile provided the most compelling evidence that this profile represented granules. The detection of further canonical P-body and

stress granules proteins with a similar profile following UPR bolstered this interpretation. Had the protein and RNA samples not been collected simultaneously, it would not have been possible to utilise the protein data to interpret the RNA profiles.

Regarding new biological analysis, the simultaneous analysis of RNA and protein localisation also allowed us to predict that binding of ZNF622 is associated with increased relocalisation of RNA to stress granules upon activation of the UPR and that ZNF622 itself relocalises away from the ER and towards the ribosomes (validated by immunofluorescence). Whilst the specific characterisation of the mechanisms underlying these findings are out of the scope of this manuscript, we believe this illustrates the benefits of studying RNA and protein localisation simultaneously.

– The criteria used to infer RNA localization to granules seem reasonable and consistent with RNA localization to granules though I am not convinced that they prove localization to granules.

We are glad that the reviewer finds our criteria to infer RNA localization to granules reasonable and consistent with RNA relocalization to granules. We would like to reiterate that, the evidence that our 'cytosol light' profile represents the correlation profile of granules is as follows:

1. Upon UPR activation, four canonical stress granule proteins (STAU2, UPF1, PABPC1 and PABPC) relocalise to the densities which discriminates cytosol light from cytosol
2. A further 7 P-body proteins and 3 stress granules proteins have highly correlated profiles with these four proteins upon UPR activation
3. Cytosolic granules containing GFP-tagged DCP2 (P-body) and G3BP1 (Stress granule) proteins are obtained within the density ranges that discriminate cytosol light from cytosol.
4. In unstressed cells, RNAs with higher cytosol light proportions have lower ribosome association and longer transcript length, features that are associated with partitioning to granules.
5. RNAs previously found to be enriched in P-bodies and TIS granules using fluorescence-activated particle sorting have a similar distribution to the cytosol light profile in unstressed conditions.
6. Upon UPR induction, the RNAs that re-localise to the cytosol light profile correlate with the RNAs previously determined to relocalise to stress granules upon another oxidative stress, arsenite treatment.

We believe this combined evidence is sufficiently strong to assert that the cytosol light profile most likely represents RNAs in granules.

– The authors seem to have collected and analyzed 20 off-line fractions but report data for only about 5314 proteins. Why is the proteome coverage not higher ?

We performed 3 replicate experiments and required the quantification of a protein in all 3 replicates for it to be reported in the results. Combined with the removal of peptide spectrum matches where > 20% of samples were not quantified and proteins with just a single peptide detected, we necessarily reported results for far fewer proteins than we detected. In total, we detected 10408 proteins over our 3 replicates (excluding common contaminants) and 6927

proteins passed the filtering in at least one replicate. 5314 proteins were quantified in all 3 replicates. While removing these filters could boost the number of proteins in our reported results, we chose to provide high-quality data for 5314 proteins.

– The biological results of the analysis are well described, though my enthusiasm for them is reduced by the use of a non physiological condition, Thapsigargin treatment, which triggers UPR and other cellular responses.

We are pleased that the reviewer thought the analysis was well described and we appreciate their reasonable concerns about the use of a pharmacological activator of the UPR, such as Thapsigargin. While we agree that a physiological strategy to activate the UPR would have provided more relevant biological insights, we selected Thapsigargin because of the following considerations. To fully test the ability of our method to capture dynamic changes in RNA and protein localisation, and to minimise confounding effects that might arise from variations in abundance, we sought a strategy that would induce robust changes very rapidly. While this is easily attainable with pharmacological inducers like Thapsigargin, physiological methods such as glucose deprivation, addition of cytokines to the media, or overexpression of mutant proteins require longer times between the start of the perturbation and a robust activation of the UPR. Moreover, while physiological strategies have been extensively used in clinically relevant contexts (e.g. macrophages and pancreatic islets); a large number of seminal UPR studies have relied on pharmacological agents as they tend to provide a more general response, i.e. they work across most cell types and activate all the three branches of the UPR, which is not always the case for physiological inducers. This was particularly important for this manuscript, as the studies used to confirm and contextualise the results from our novel method all use pharmacological methods to induce the UPR (e.g. Thapsigargin in^{11,12}; DTT in¹³). Therefore, using a physiological activator of the unfolded protein response could have severely complicated any comparison with these studies, which is a fundamental aspect of the manuscript.

Reviewer #3:

Remarks to the Author:

In this manuscript, the authors present an integrated approach where they combine localization of RNAs to a density-based localization approach. The novelty of this method is that the authors study the subcellular structures by their density and quantify RNA abundance profiles across the density fractions with proteomic measurements made in parallel. They also apply this to study the redistribution of RNA and protein during the activation of UPR. The authors present some interesting findings related to localization of lncRNAs, discovery of membrane-localized RNAs lacking signal peptide or TM domains and recruitment of ER-localized RNAs to granules. I feel that the manuscript is acceptable for publication after the following issues are addressed:

1. The dLOPIT method is a modification to a previously published method (Localization of

organelle proteins by isotope tagging (LOPIT)) by the group in 2004 (Mol Cell Proteomics. 2004.3(11):1128-34) but seems to be highlighted here as a completely new method.

We did not mean to imply that dLOPIT is a completely new method. Indeed, we kept the LOPIT acronym to indicate that this method builds on the knowledge we have accumulated with the previous protein subcellular localisation methods that we have developed to address this specific biological question. Nevertheless, dLOPIT represents a paradigm shift to our previous LOPITs, making it, for the first time, compatible with the simultaneous determination of the subcellular localisation of another macromolecule. We have made clearer, both in the introduction and in the results, that dLOPIT is a development of the principles of correlation profiling that we previously applied in LOPIT.

2. The discussion about lncRNAs is not clear. I believe that the authors should have data that allows them to correlate their localization findings with several other features:
 i) Are they able to discover lncRNAs that have been shown to encode proteins based on their data?
 ii) Can they correlate the proteomic data with translation of the subset of lncRNAs that encode proteins?

Figure S3C shows how the cytosolic proportion for lncRNAs relates to their ribosome association. As expected, we observed a positive correlation between ribosome association and cytosolic proportions, meaning that cytosolic lncRNAs are more likely to be associated with ribosomes than the nuclear-retained lncRNAs. Nevertheless, ribosome association does not directly imply translation.

We agree it would be interesting to use LoRNA to interrogate the subcellular localisation of lncRNA translation products. Unfortunately, we utilised Real Time Search during the acquisition of our quantitative proteomics data to maximise the number of quantified peptides. This means we cannot re-search our Mass Spectrometry data to quantify further peptides that were not in our reference at the time of data acquisition and we did not include putative lncRNA peptides in our original reference.

3. Overall, the manuscript is hard to read. It would be great if the authors could simplify the manuscript to make it more accessible to a wide audience.

We hope that the reviewer finds the manuscript easier to read now we have addressed the comments from all three reviewers.

4. The authors do not discuss any data pertaining to mitochondria in the text. Is this because of the overlap with ER? I believe mitochondria would serve as a nice control because of mitochondrial transcription, RNA processing and translation within the organelle and it would be nice to know if the authors found expected correlations in their data related to mitochondria.

We agree with the reviewer that the mitochondria represent the ideal control for membrane-associated RNAs since mitochondrial-genome-encoded RNAs are the only expected transcripts in the cell to have a 100% organelle association. This is especially true when interrogating RNA dynamics in conditions such as UPR induction where ER-translated RNAs lose their ER associated. Indeed, this is the reason why we have used the mitochondrial-

encoded RNAs (Fig 1d and Fig 2d), as markers for 100% membrane association. This allowed us to quantify RNA relocalisation out of the ER upon UPR induction. We have added the following text to the results to clarify this: *'Importantly, we took advantage of the extreme localisation of the 13 mitochondrially-encoded mRNAs to accurately quantify membrane association of RNA, using these RNAs as a reference for 100% membrane localisation'*

5. I am not convinced that this approach deserves another acronym: Subcellular Multiomics Mapping (SuMM). Another reason is that a lot of conclusions are based on correlations and not direct observations/measurements although the term 'mapping' implies greater precision. We agree with the reviewer that the term mapping can be confusing for some readers that may expect mapping to spatial coordinates instead of mapping to subcellular compartments. We have amended the manuscript accordingly and now refer to a 'new integrative framework' within giving it a new acronym.

References:

1. de Duve, C. Tissue fraction-past and present. *J. Cell Biol.* **50**, 20 (1971).
2. Foster, L. J. *et al.* A Mammalian Organelle Map by Protein Correlation Profiling. *Cell* **125**, 187–199 (2006).
3. Xia, C., Fan, J., Emanuel, G., Hao, J. & Zhuang, X. Spatial transcriptome profiling by MERFISH reveals subcellular RNA compartmentalization and cell cycle-dependent gene expression. *Proc. Natl. Acad. Sci.* **116**, 19490–19499 (2019).
4. Thul, P. J. *et al.* A subcellular map of the human proteome. *Science* **356**, eaal3321 (2017).
5. Geladaki, A. *et al.* Combining LOPIT with differential ultracentrifugation for high-resolution spatial proteomics. *Nat. Commun.* **10**, 331 (2019).
6. Christoforou, A. *et al.* A draft map of the mouse pluripotent stem cell spatial proteome. *Nat. Commun.* **7**, 9992 (2016).
7. Benoit Bouvrette, L. P. *et al.* CeFra-seq reveals broad asymmetric mRNA and noncoding RNA distribution profiles in *Drosophila* and human cells. *RNA* **24**, 98–113 (2018).
8. Wang, E. T. *et al.* Transcriptome-wide Regulation of Pre-mRNA Splicing and mRNA Localization by Muscleblind Proteins. *Cell* **150**, 710–724 (2012).
9. Bhatt, D. M. *et al.* Transcript Dynamics of Proinflammatory Genes Revealed by Sequence Analysis of Subcellular RNA Fractions. *Cell* **150**, 279–290 (2012).
10. Adekunle, D. A. & Wang, E. T. Transcriptome-wide organization of subcellular microenvironments revealed by ATLAS-Seq. *Nucleic Acids Res.* **48**, 5859–5872 (2020).
11. Guan, B.-J. *et al.* A Unique ISR Program Determines Cellular Responses to Chronic Stress. *Mol. Cell* **68**, 885–900.e6 (2017).
12. Shaban, M. S. *et al.* Multi-level inhibition of coronavirus replication by chemical ER stress. *Nat. Commun.* **12**, 5536 (2021).
13. Child, J. R., Chen, Q., Reid, D. W., Jagannathan, S. & Nicchitta, C. V. Recruitment of endoplasmic reticulum-targeted and cytosolic mRNAs into membrane-associated stress granules. *RNA N. Y. N* **27**, 1241–1256 (2021).

Decision Letter, first revision:

Dear Kathryn,

Thank you for submitting your revised manuscript "System-wide analysis of RNA and protein subcellular localisation dynamics" (N METH-A51585A). It has now been seen by the original referees and their comments are below. The reviewers find that the paper has improved in revision, and therefore we'll be happy in principle to publish it in Nature Methods, pending minor revisions to satisfy the referees' final requests and to comply with our editorial and formatting guidelines.

TRANSPARENT PEER REVIEW

Nature Methods offers a transparent peer review option for new original research manuscripts submitted from 17th February 2021. We encourage increased transparency in peer review by publishing the reviewer comments, author rebuttal letters and editorial decision letters if the authors agree. Such peer review material is made available as a supplementary peer review file. Please state in the cover letter 'I wish to participate in transparent peer review' if you want to opt in, or 'I do not wish to participate in transparent peer review' if you don't. Failure to state your preference will result in delays in accepting your manuscript for publication.

ORCID

Sincerely,
Arunima

Arunima Singh, Ph.D.
Senior Editor
Nature Methods

Reviewer #1 (Remarks to the Author):

The revised manuscript presents significantly improved data analysis and in particular the comparison with data generated from totally different techniques that show reasonably good concordance. Specifically, the comparison with MERFISH data is quite compelling. The comparison with Human Protein Atlas data shows something interesting as well. Overall, the manuscript has been much improved and this reviewer has no more major concerns.

Reviewer #2 (Remarks to the Author):

Some of the author responses appear to sidestep my questions, but as a whole I find that the responses satisfactorily address core concerns and the revisions improved the paper.

I found the added benchmarking in response to Rev # 1 useful, but I was not sure how to interpret 50 - 59 % agreement with the Human Protein Atlas (HPA). It would be useful to compare this agreement to the agreement for a reasonable null model, e.g., randomly distributed proteins across compartments with the number of proteins in each compartment corresponding to the observed number.

I do not need to review the results from such a null model -- it's just a suggestion that the authors may find helpful and use to strengthen their claim about close agreement.

Reviewer #3 (Remarks to the Author):

My concerns have been addressed by the authors. I feel that the quality of this manuscript has been improved in response to all of the reviewers' comments.

Author Rebuttal, first revision:

Reviewer #1:

Remarks to the Author:

The revised manuscript presents significantly improved data analysis and in particular the comparison with data generated from totally different techniques that show reasonably good concordance. Specifically, the comparison with MERFISH data is quite compelling. The comparison with Human Protein Atlas data shows something interesting as well. Overall, the manuscript has been much improved and this reviewer has no more major concerns.

Reviewer #2:

Remarks to the Author:

Some of the author responses appear to sidestep my questions, but as a whole I find that the responses satisfactory address core concerns and the revisions improved the paper.

I found the added benchmarking in response to Rev # 1 useful, but I was not sure how to interpret 50 - 59 % agreement with the Human Protein Atlas (HPA). It would be useful to compare this agreement to the agreement for a reasonable null model, e.g., randomly distributed proteins across compartments with the number of proteins in each compartment corresponding to the observed number.

I do not need to review the results from such a null model -- it's just a suggestion that the authors may find helpful and use to strengthen their claim about close agreement.

We thank the reviewer for the suggestion to include a null model for comparison. As suggested, to obtain a null distribution, we have sampled the compartments randomly, with the same probability for each compartment as the observed frequencies with each LOPIT. We have also refined the comparison so that we are comparing at each individual level of confidence for the HPA annotations, to simplify the interpretation. Previously, we were considering all annotations above and including a given confidence level. In all cases, the observed agreement with HPA is outside the null distribution from 1000 iterations. The new analysis is included in supplementary figure 1h.

Reviewer #3:

Remarks to the Author:

My concerns have been addressed by the authors. I feel that the quality of this manuscript has been improved in response to all of the reviewers' comments.

Final Decision Letter:

Dear Kathryn,

I am pleased to inform you that your Article, "System-wide analysis of RNA and protein subcellular localisation dynamics", has now been accepted for publication in Nature Methods. Your paper is tentatively scheduled for publication in our January print issue, and will be published online prior to that. The received and accepted dates will be January 27, 2023 and October 24, 2023. This note is intended to let you know what to expect from us over the next month or so, and to let you know where to address any further questions.

Over the next few weeks, your paper will be copyedited to ensure that it conforms to Nature Methods style. Once your paper is typeset, you will receive an email with a link to choose the appropriate publishing options for your paper and our Author Services team will be in touch regarding any additional information that may be required.

You will receive a link to your electronic proof via email with a request to make any corrections within 48 hours. If, when you receive your proof, you cannot meet this deadline, please inform us at rjsproduction@springernature.com immediately.

Please note that *Nature Methods* is a Transformative Journal (TJ). Authors may publish their research with us through the traditional subscription access route or make their paper immediately open access through payment of an article-processing charge (APC). Authors will not be required to make a final decision about access to their article until it has been accepted. [Find out more about Transformative Journals](https://www.springernature.com/gp/open-research/transformative-journals)

Your paper will now be copyedited to ensure that it conforms to Nature Methods style. Once proofs are generated, they will be sent to you electronically and you will be asked to send a corrected version within 24 hours. It is extremely important that you let us know now whether you will be difficult to contact over the next month. If this is the case, we ask that you send us the contact information (email, phone and fax) of someone who will be able to check the proofs and deal with any last-minute problems.

If, when you receive your proof, you cannot meet the deadline, please inform us at rjsproduction@springernature.com immediately.

Once your manuscript is typeset and you have completed the appropriate grant of rights, you will receive a link to your electronic proof via email with a request to make any corrections within 48 hours. If, when you receive your proof, you cannot meet this deadline, please inform us at rjsproduction@springernature.com immediately.

Once your paper has been scheduled for online publication, the Nature press office will be in touch to confirm the details.

Once your paper has been scheduled for online publication, the Nature press office will be in touch to confirm the details.

Content is published online weekly on Mondays and Thursdays, and the embargo is set at 16:00 London time (GMT)/11:00 am US Eastern time (EST) on the day of publication. If you need to know the exact publication date or when the news embargo will be lifted, please contact our press office after you have submitted your proof corrections. Now is the time to inform your Public Relations or Press Office about your paper, as they might be interested in promoting its publication. This will allow them time to prepare an accurate and satisfactory press release. Include your manuscript tracking number NMETH-A51585B and the name of the journal, which they will need when they contact our office.

About one week before your paper is published online, we shall be distributing a press release to news organizations worldwide, which may include details of your work. We are happy for your institution or funding agency to prepare its own press release, but it must mention the embargo date and Nature Methods. Our Press Office will contact you closer to the time of publication, but if you or your Press Office have any inquiries in the meantime, please contact press@nature.com.

Nature Portfolio journals [encourage authors to share their step-by-step experimental protocols](https://www.nature.com/nature-research/editorial-policies/reporting-standards#protocols) on a protocol sharing platform of their choice. Nature Portfolio 's Protocol Exchange is a free-to-use and open resource for protocols; protocols deposited in Protocol Exchange are citable and can be linked from the published article. More details can found at www.nature.com/protocolexchange/about.

Best regards,
Arunima

Arunima Singh, Ph.D.
Senior Editor
Nature Methods